# MoCo-EA: Exploiting Adversarial Mode Connectivity for Efficient Evolutionary Attacks

## Abstract

Evolutionary algorithms for adversarial attacks leverage population-based search to discover perturbations without gradient information, but suffer from inefficient crossover operations that destroy adversarial properties through discrete interpolation. We introduce Mode Connectivity Evolutionary Attack (MoCo-EA), which replaces traditional crossover with a novel Bézier crossover operator that optimizes perturbations along a continuous Bézier curve between parent perturbations. Our key insight is that adversarial examples lie on connected manifolds where intermediate points maintain, and often enhance attack effectiveness. We demonstrate three findings: (1) Successful adversarial perturbations exhibit mode connectivity, forming continuous paths that preserve adversarial properties; (2) Intermediate points along optimized paths achieve higher transferability than endpoints, with improvements that scale with auxiliary image guidance; (3) Bézier crossover dramatically outperforms discrete genetic operations, achieving universal attack success across all perturbation norms while reducing convergence time and query requirements by orders of magnitude. By revealing the geometric structure of adversarial space and exploiting it through principled path optimization, MoCo-EA transforms evolutionary attacks from slow and unreliable processes into efficient and dependable methods. Our work challenges the traditional view of adversarial examples as isolated points and opens new directions for both attack generation and defense research.

## 1 Introduction

Adversarial attacks expose the vulnerability of deep neural networks to carefully crafted input perturbations (Goodfellow et al., 2015; Madry et al., 2018). These attacks are broadly categorized as white-box or black-box, depending on the attacker's access to the model (Costa et al., 2024). White-box attacks leverage full model access, including gradients, enabling effective methods such as Fast Gradient Sign Method (FGSM) (Goodfellow et al., 2015), Projected Gradient Descent (PGD) (Madry et al., 2018), C&W (Carlini & Wagner, 2017), AutoAttack (Croce & Hein, 2020), and Deep-Fool (Moosavi-Dezfooli et al., 2016). These optimize perturbations under $\ell_p$-norm constraints and remain state-of-the-art in bounded settings. In contrast, black-box attacks rely on model queries or surrogates, including score-based (Chen et al., 2017), decision-based (Chen et al., 2020; Brendel et al., 2018), and evolutionary methods like GenAttack (Alzantot et al., 2019) and the one-pixel attack (Su et al., 2019). Among these, evolutionary algorithms constitute a distinct category that operates without gradient information, offering inherent parallelizability for effective exploration and population diversity to escape local optima. They typically evolve a population of perturbations using operators like mutation, selection, and crossover, mimicking biological evolution.

However, evolutionary attacks are rarely explored in the white-box setting. This is surprising, since white-box access provides valuable signals that could inform and improve population-based search. Existing methods like GenAttack are purely gradient-free and use element-wise crossover operations that ignore the underlying geometry of the input space. As a result, they tend to suffer from inefficiencies, poor transferability, and limited diversity.

To address these limitations, we discover and exploit a previously unexplored property: adversarial mode connectivity (the existence of continuous paths between different adversarial perturbations that maintain attack effectiveness throughout). While mode connectivity has been extensively studied in

neural network parameter spaces (Garipov et al., 2018; Draxler et al., 2018; Freeman & Bruna, 2017) and recently extended to functional and permutation spaces (Zhao et al., 2020; Entezari et al., 2022), its application to bridging adversarial perturbations remains unexplored. We demonstrate that successful adversarial examples lie on connected manifolds where continuous paths preserve adversarial properties. More importantly, we find that intermediate points along optimized paths exhibit significantly higher transferability than endpoints, with substantial improvements in attack success and rescue rates for previously failed attacks. This discovery reveals that the adversarial space has rich geometric structure amenable to continuous exploration rather than discrete sampling. Related advances in input-space connectivity further highlight this potential, as shown by Vrabel et al. (2025) and Kariyappa & Qureshi (2019).

Building on these insights, we propose Mode Connectivity Evolutionary Attack (MoCo-EA), which fundamentally reimagines crossover through continuous path optimization. Our approach systematically studies adversarial perturbation connectivity, showing that successful attacks are not isolated points but lie on connected adversarial manifolds. We further reveal that intermediate points on optimized Bézier curves achieve stronger and more transferable attacks than endpoints. Finally, we develop MoCo-EA, replacing discrete crossover with Bézier curve interpolation, achieving universal attack success while sharply reducing convergence time and query requirements. An overview of the proposed MoCo-EA is shown in Fig. 1. In brief, from two parent perturbations, we optimize a quadratic Bézier path in perturbation space to connect those parents, evaluate the resulting connectivity under progressively harder settings and with multi-image augmentation, and finally instantiate a geometry-aware evolutionary attack that employs a Bézier crossover operator.

We summarize our contributions below:

- We provide the first systematic study of adversarial perturbation connectivity, showing that successful attacks are not isolated points but are connected by low-loss paths with preserved adversarial properties.

- We reveal that intermediate points on optimized Bézier paths between parents are stronger than endpoints, with transferability that increases monotonically as more auxiliary images guide path optimization.

- We develop MoCo-EA, a geometry-aware evolutionary attack that replaces discrete crossover with Bézier crossover, achieving near-perfect success across norms while sharply cutting convergence time and query complexity.

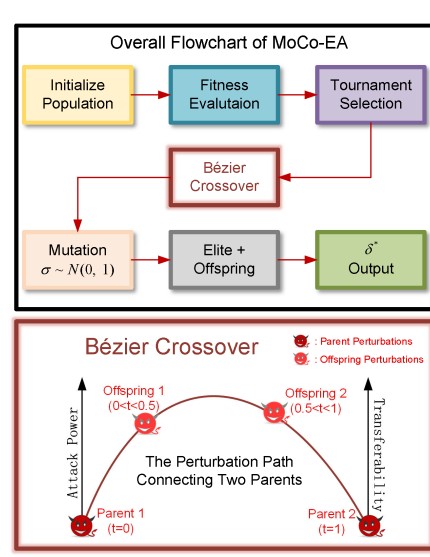

Figure 1: Overview of MoCo-EA.

## 2 RELATED WORK

**Gradient-based adversarial attacks.** Gradient-based attacks are the most widely used methods in the white-box setting, where the attacker has full access to model parameters and gradients. The single-step FGSM introduced by Goodfellow et al. (2015) and its multi-step variant, PGD (Madry et al., 2018), became standard white-box attacks and the backbone of adversarial training. Optimization-based C&W attacks target different $\ell_p$ norms (Carlini & Wagner, 2017), while AutoAttack provides a robust ensemble for evaluation (Croce & Hein, 2020). Stronger transfer-based variants incorporate momentum (Dong et al., 2018), input diversity (Xie et al., 2019), and translation invariance (Dong et al., 2019). However, most gradient-based methods operate locally around a single example and do not explicitly leverage global structure that might connect different adversarial modes, motivating our study of continuous connectivity between successful perturbations.

**Evolutionary adversarial attacks.** Evolutionary algorithms have proven effective in black-box scenarios due to their ability to explore complex and non-differentiable search spaces. Representative methods include GenAttack (Alzantot et al., 2019) and the one-pixel differential-evolution

attack (Su et al., 2019). Subsequent work improved sampling/guidance via probability-guided genetic search (Li et al., 2020) and gradient/score estimation (Wang et al., 2021). However, they remain underexplored in white-box settings, where gradient information could enhance evolutionary dynamics. Most existing evolutionary attacks use element-wise crossover and heuristic mutations, which are agnostic to the structure of the data manifold or loss surface. Moreover, simply incorporating gradients into mutation steps or fitness scoring may lead to instability or mode collapse. Our work departs from these approaches by proposing a structured and geometry-aware evolutionary attack that explicitly models the connectivity between adversarial examples. Unlike prior methods that treat perturbations as isolated points in input space, we introduce Bézier-based crossover that leverages gradient-informed path optimization to interpolate through adversarial modes. This enables us to preserve adversarial properties along the path while improving sample diversity and transferability.

## 3 ADVERSARIAL MODE CONNECTIVITY

### 3.1 BÉZIER CURVE FOR MODE CONNECTIVITY

Our approach begins by identifying two distinct adversarial perturbations that serve as endpoints for our Bézier curve construction. We employ PGD (Madry et al., 2018), which remains one of the strongest first-order adversarial attack methods. Given a clean image $x \in [0, 1]^d$ with true label $y$, and a classifier $f_{\theta}$, PGD solves:

$$\max_{\|\delta\|_p \leq \epsilon} \mathcal{L}(f_{\theta}(x + \delta), y), \tag{1}$$

where $\mathcal{L}$ is the cross-entropy loss, and $\| \cdot \|_p$ constrains the perturbation within an $\ell_p$-ball of radius $\epsilon$. To obtain two distinct local minima $\delta_1$ and $\delta_2$, we run PGD twice with different random initializations; each run starts from a perturbation sampled uniformly from $(-\epsilon, \epsilon)$ using different seeds. Both $\delta_1$ and $\delta_2$ fool the classifier (i.e., $f_{\theta}(x + \delta_1) \neq y$ and $f_{\theta}(x + \delta_2) \neq y$) while capturing different adversarial patterns in the perturbation space.

Adversarial solutions obtained from different initializations often correspond to different local optima, yet they can be connected in the adversarial perturbation space by a path along which predictions remain adversarial. We use mode connectivity to explicitly search such a path because it exposes shared structure among adversarial solutions and helps maintain adversarial effectiveness along the whole trajectory. Among possible parameterizations, we adopt a quadratic Bézier curve, which is widely used in mode-connectivity settings and provides an efficient and effective two-endpoint parameterization with a single learnable control point. Quadratic Bézier curves also offer a favorable expressivity–stability trade-off: they are flexible enough to capture curved low-loss connections while remaining amenable to stable optimization and norm-ball projection. In contrast to discrete element-wise crossover, which often disrupts adversarial structure, Bézier crossover preserves coherence by construction and can be optimized with only a few gradient steps.

Given two adversarial endpoints $\delta_1$ and $\delta_2$, we define the quadratic Bézier curve as follows:

$$\mathbf{B}(t; \delta_c) = (1 - t)^2 \delta_1 + 2(1 - t)t \, \delta_c + t^2 \delta_2, \quad t \in [0, 1], \tag{2}$$

where $\delta_c$ is the learnable control point that determines the curvature of the path.

We initialize $\delta_c^{(0)} = \frac{1}{2}(\delta_1 + \delta_2)$ and optimize it by maximizing adversarial loss along the curve:

$$\delta_c^* = \arg\min_{\delta_c} \mathbb{E}_{t \sim \mathcal{U}(0,1)} \Big[ - \mathcal{L}\big(f_{\theta}\big(x + \Pi_{\|\cdot\|_p \leq \epsilon} [\mathbf{B}(t; \delta_c)]\big), y\big) \Big], \tag{3}$$

where $\Pi$ denotes projection onto the $\ell_p$-ball of radius $\epsilon$, and $t$ is sampled from the uniform distribution $\mathcal{U}(0, 1)$ during optimization.

This framework allows us to optimize entire perturbation paths rather than isolated endpoints, enabling a direct examination of the structure of adversarial perturbations in subsequent sections. Classical mode-connectivity studies show that distinct solutions can often be linked by smooth paths that preserve low loss(Garipov et al., 2018). In our setting, we extend this structural idea to the space of adversarial perturbations, but with the objective inverted. Rather than maintaining low loss, we search for continuous trajectories along which the adversarial loss remains high. A geometric intuition in ReLU network is discussed in Appendix A.2.

## 3.2 CONNECTIVITY SETTINGS: FROM IMAGE-WISE TO CROSS-CLASS

PGD optimizes a single perturbation for one image and therefore tends to converge to sharp, highly localized adversarial maxima with limited transferability(Qin et al., 2022). In contrast, optimizing the Bézier control point requires maintaining high loss at multiple sampled points along the curve, which prevents the solution from collapsing onto such sharp maxima. This multi-point objective encourages the entire trajectory to move toward flatter and more stable high-loss regions, where perturbations generally exhibit stronger cross-instance generalization.

We adopt three settings to systematically study adversarial mode connectivity under different levels of generalization. Setting A focuses on a single image, ensuring the feasibility of connecting two adversarial modes in the simplest case. Setting B extends to multiple images of the same class, testing whether a single curve parameter $\boldsymbol{\delta}_c$ can generalize across variations in appearance while keeping the label fixed. Setting C considers images from different classes, which is the most challenging scenario due to greater semantic dissimilarity.

**Setting A (Image-wise Connectivity).** For a single image $\boldsymbol{x}$ with label $\boldsymbol{y}$, we find $\boldsymbol{\delta}_1, \boldsymbol{\delta}_2$ via PGD on the same image and optimize

$$\boldsymbol{\delta}_c^* = \arg\min_{\boldsymbol{\delta}_c} \; \mathbb{E}_{t\sim\mathcal{U}(0,1)}\Big[ - \mathcal{L}\big(f_{\boldsymbol{\theta}}(\boldsymbol{x} + \Pi_{\|\cdot\|_p\leq\epsilon}[\mathbf{B}(t;\boldsymbol{\delta}_c)]), \boldsymbol{y}\big)\Big]. \tag{4}$$

This setting evaluates whether adversarial modes for a single image can be connected while still maintaining attack success.

**Setting B (Class-wise Connectivity).** Given two images $\boldsymbol{x}_1, \boldsymbol{x}_2$ from the same class $\boldsymbol{y}$, we compute $\boldsymbol{\delta}_1 = \mathrm{PGD}(\boldsymbol{x}_1, \boldsymbol{y})$ and $\boldsymbol{\delta}_2 = \mathrm{PGD}(\boldsymbol{x}_2, \boldsymbol{y})$, then optimize

$$\boldsymbol{\delta}_c^* = \arg\min_{\boldsymbol{\delta}_c} \; \frac{1}{2}\sum_{i=1}^{2} \mathbb{E}_{t\sim\mathcal{U}(0,1)}\Big[ - \mathcal{L}\big(f_{\boldsymbol{\theta}}(\boldsymbol{x}_i + \Pi_{\|\cdot\|_p\leq\epsilon}[\mathbf{B}(t;\boldsymbol{\delta}_c)]), \boldsymbol{y}\big)\Big]. \tag{5}$$

This setting evaluates whether adversarial perturbations discovered on different samples of the same class can be connected in a unified curve.

**Setting C (Cross-class Connectivity).** For images $\boldsymbol{x}_1, \boldsymbol{x}_2$ from different classes $\boldsymbol{y}_1, \boldsymbol{y}_2$, with $\boldsymbol{\delta}_1 = \mathrm{PGD}(\boldsymbol{x}_1, \boldsymbol{y}_1)$ and $\boldsymbol{\delta}_2 = \mathrm{PGD}(\boldsymbol{x}_2, \boldsymbol{y}_2)$, we optimize

$$\boldsymbol{\delta}_c^* = \arg\min_{\boldsymbol{\delta}_c} \; \frac{1}{2}\sum_{i=1}^{2} \mathbb{E}_{t\sim\mathcal{U}(0,1)}\Big[ - \mathcal{L}\big(f_{\boldsymbol{\theta}}(\boldsymbol{x}_i + \Pi_{\|\cdot\|_p\leq\epsilon}[\mathbf{B}(t;\boldsymbol{\delta}_c)]), \boldsymbol{y}_i\big)\Big]. \tag{6}$$

This setting examines whether adversarial connectivity can hold across different semantic classes, which is the most general and challenging scenario.

## 3.3 MULTI-IMAGE AUGMENTATION FOR ENHANCED TRANSFERABILITY

To improve the transferability of discovered adversarial curves, we use two kinds of images during optimization. The main images are those used to define the endpoints of the curve. Auxiliary images are additional samples that regularize and improve the transferability of the learned curve. These auxiliary images encourage the curve to encode perturbations that generalize across visual variations. For Setting A we select auxiliary images from the same class as $\boldsymbol{y}$; for Setting B from the same class $\boldsymbol{y}$; for Setting C we select a balanced set from the two classes $\boldsymbol{y}_1$ and $\boldsymbol{y}_2$. In all cases, auxiliary images are drawn from a held-out pool distinct from training and test splits.

For any image $(\boldsymbol{x}_k, \boldsymbol{y}_k)$ (main or auxiliary), the per-image adversarial loss along the curve follows the same construction as in equation 3:

$$\mathcal{L}_k(t;\boldsymbol{\delta}_c) := \mathcal{L}\big(f_{\boldsymbol{\theta}}\big(\boldsymbol{x}_k + \Pi_{\|\cdot\|_p\leq\epsilon}[\mathbf{B}(t;\boldsymbol{\delta}_c)]\big), \boldsymbol{y}_k\big), \tag{7}$$

where $t \sim \mathcal{U}(0,1)$ and $\mathbf{B}(t;\boldsymbol{\delta}_c)$ is the quadratic Bézier path shared across all images.

Given nonnegative weights $w_{\text{main}}$ and $w_{\text{aux}}$ ($w_{\text{main}} > w_{\text{aux}}$ to emphasize the main images), we optimize

$$\boldsymbol{\delta}_c^* = \arg\min_{\boldsymbol{\delta}_c} \; \mathbb{E}_{t \sim \mathcal{U}(0,1)} \left[ -\sum_{i \in \text{main}} w_{\text{main}} \, \mathcal{L}_i^{\text{main}}(t; \boldsymbol{\delta}_c) - \sum_{j \in \text{aux}} w_{\text{aux}} \, \mathcal{L}_j^{\text{aux}}(t; \boldsymbol{\delta}_c) \right], \qquad (8)$$

which maximizes adversarial loss along the curve for both main and auxiliary images while respecting the $\ell_p$-budget via the projection operator.

When auxiliary images are incorporated, the control point $\boldsymbol{\delta}_c$ must further induce high loss across multiple inputs at once. This multi-image objective acts as an implicit regularizer that biases the optimization toward perturbation directions that remain adversarial under larger input variation. As a result, the learned path increasingly aligns with more universal high-loss directions, providing a natural explanation for the strong transferability.

## 4 MODE CONNECTIVITY EVOLUTIONARY ATTACK (MOCO-EA)

### 4.1 TRADITIONAL EVOLUTIONARY ALGORITHMS FOR ADVERSARIAL ATTACKS

We assume a white-box threat model with full knowledge of the model parameters. In this white-box setting we can utilize adversarial mode connectivity to replace the crossover operation used in traditional evolutionary attacks.

Evolutionary algorithms (EAs) have emerged as powerful gradient-free methods for generating adversarial examples, benefiting from population diversity to escape local optima. The traditional EA framework for adversarial attacks operates through iterative population-based optimization. We initialize a population $P^{(0)} = \{\boldsymbol{\delta}_1, \boldsymbol{\delta}_2, \ldots, \boldsymbol{\delta}_N\}$ of $N$ random perturbations within the $\epsilon$-ball, evaluate a fitness score for each perturbation based on attack success, and use tournament selection to choose parent pairs for reproduction. Traditional crossover combines two parent perturbations element-wise, and mutation is implemented by adding Gaussian noise with some probability $p_m$:

$$\text{child}[j] = \begin{cases} \text{parent}_1[j] & \text{with } \Pr(0.5), \\ \text{parent}_2[j] & \text{otherwise.} \end{cases} \quad \text{(Crossover)}, \quad \boldsymbol{\delta}' = \boldsymbol{\delta} + \eta \cdot \mathcal{N}(0, \sigma^2 I) \quad \text{(Mutation)}. \quad (9)$$

Here, $\eta > 0$ is the mutation step size and the mutation operator is applied to each individual with probability $p_m$. Then elite preservation retains the top-$k$ individuals for the next generation. The fundamental limitation of traditional crossover is that its discrete, element-wise mixing tends to break spatial and structural coherence of successful adversarial patterns. Randomly combining pixels or features from two strong parents can produce offspring that no longer fool the classifier; moreover, uniform crossover is agnostic to the loss landscape between parents and may create children that lie in regions of low adversarial effectiveness. Finally, the element-wise nature restricts the search to certain combinations of parent features and can limit exploration of more structured low-loss paths between adversarial modes.

### 4.2 MOCO-EA ALGORITHM OVERVIEW

We propose MoCo-EA, which enhances traditional evolutionary algorithms by replacing the traditional discrete crossover operator with a geometry-aware Bézier crossover. The overall algorithm maintains a population of candidate perturbations and evolves them through selection, Bézier crossover, and mutation, following the general structure of EAs but innovating in its crossover mechanism. The Bézier crossover operator is detailed below in Algorithm 1. For the complete MoCo-EA procedure, please refer to A.3 (Algorithm 2). An overview of the pipeline is shown in Fig. 1.

The procedure begins by initializing a population of $N$ random perturbations inside the $\ell_p$ $\epsilon$-ball around the input. Each perturbation is evaluated with a fitness function based on its ability to cause misclassification. In each generation, parent pairs are selected from the population according to their fitness. The Bézier crossover operator then takes two parents, $\delta_1$ and $\delta_2$, and connects them with a quadratic Bézier curve parameterized by a control point $\boldsymbol{\delta}_c$. The control point $\boldsymbol{\delta}_c$ is optimized for a few gradient steps to maximize adversarial loss along sampled points on the path, with each point projected back to the $\epsilon$-ball to satisfy the perturbation constraint. After this optimization, new

offspring are generated by sampling from different regions of the curve. Points closer to $\delta_1$ and $\delta_2$ are used to form distinct children, and among multiple samples the highest-fitness ones are chosen. Each selected offspring is projected back to the feasible set.

---

**Algorithm 1** Bézier Crossover

---

1: **Input:** parent perturbations $\boldsymbol{\delta}_1, \boldsymbol{\delta}_2$; image $\boldsymbol{x}$; label $y$; model $f_{\boldsymbol{\theta}}$
2: **Parameters:** control-step count $\tau$, step size $\alpha$, projection $\Pi_{\|\cdot\|_p \leq \epsilon}$
3: **Output:** two offspring perturbations
4: $\boldsymbol{\delta}_c \leftarrow (\boldsymbol{\delta}_1 + \boldsymbol{\delta}_2)/2$
5: **for** $step = 1$ **to** $\tau$ **do**
6: $\quad loss \leftarrow 0$
7: $\quad$ **for** $t \in \{0.25, 0.5, 0.75\}$ **do**
8: $\quad\quad \boldsymbol{\delta}_t \leftarrow \mathbf{B}(t; \boldsymbol{\delta}_c, \boldsymbol{\delta}_1, \boldsymbol{\delta}_2)$
9: $\quad\quad loss \leftarrow loss - \mathcal{L}\big(f_{\boldsymbol{\theta}}(\boldsymbol{x} + \Pi_{\|\cdot\|_p \leq \epsilon}[\boldsymbol{\delta}_t]), y\big)$
10: $\quad$ **end for**
11: $\quad \boldsymbol{\delta}_c \leftarrow \boldsymbol{\delta}_c - \alpha \cdot \nabla_{\boldsymbol{\delta}_c} loss$
12: **end for**
13: $c_1 \leftarrow \text{SelectBest}\big(\{\mathbf{B}(t; \boldsymbol{\delta}_c) \mid t \in (0, 0.5)\}\big)$
14: $c_2 \leftarrow \text{SelectBest}\big(\{\mathbf{B}(t; \boldsymbol{\delta}_c) \mid t \in (0.5, 1)\}\big)$
15: **return** $\Pi_{\|\cdot\|_p \leq \epsilon}[c_1], \Pi_{\|\cdot\|_p \leq \epsilon}[c_2]$

---

Mutation is applied to offspring with a certain probability. Elitism ensures that the top $k$ individuals from the current population are preserved. The new generation is then formed from the elites and the best offspring from crossover and mutation. This process repeats until either a successful adversarial perturbation is found or the maximum number of generations $G$ is reached.

Such trajectories instantiate adversarial mode connectivity and exhibit two important properties. First, connectivity: adversarial effectiveness is preserved along the path. Second, transferability: intermediate points on the curve often transfer better than the endpoints. Together, these properties position Bézier connectivity as an effective paradigm for structuring adversarial perturbations and provide a direct rationale for its use within an evolutionary search framework.

## 5 EXPERIMENTS

### 5.1 EXPERIMENTAL SETUP

**Dataset and Model.** We evaluate on CIFAR-10 (Krizhevsky, 2009) with a ResNet-18 (He et al., 2016) classifier, and on ImageNet (Deng et al., 2009) with a ViT-Base/16 (Dosovitskiy et al., 2021) classifier. For detailed dataset and model introduction, please review A.4.

**Adversarial Attack Settings.** On CIFAR-10, we use $\ell_\infty$ with $\epsilon = 8/255$, $\ell_2$ with $\epsilon = 0.5$, and $\ell_1$ with $\epsilon = 10$. On ImageNet, we use $\ell_\infty$ with $\epsilon = 4/255$, $\ell_2$ with $\epsilon = 2$, and $\ell_1$ with $\epsilon = 75$. For generating adversarial endpoints, we employ Projected Gradient Descent (PGD) (Madry et al., 2018) with 40 iterations using step sizes $\alpha = \epsilon/4$ for $\ell_\infty$, $\alpha = \epsilon/5$ for $\ell_2$, and $\alpha = \epsilon/10$ for $\ell_1$. For image selection protocols for Settings A/B/C, please review A.4.

**Bézier Optimization.** The control point $\delta_c$ is optimized using the Adam optimizer (Kingma & Ba, 2015) with a learning rate of 0.01 for 30 iterations. We sample 20 random $t$ values per iteration during optimization, and evaluate the final curve on 50 evenly spaced $t$ values (excluding endpoints) to compute success rates. For MoCo-EA crossover, we use a reduced 5 iterations with 3 fixed sampling points ($t \in \{0.25, 0.5, 0.75\}$) for efficiency.

### 5.2 ADVERSARIAL MODE CONNECTIVITY

**Connectivity Analysis.** We test whether continuous paths between successful adversarial perturbations preserve attack effectiveness. We evaluate attack success along optimized Bézier paths with 25 samples per setting. For each setting we (i) obtain two adversarial endpoints with PGD, (ii) build a quadratic Bézier curve $B(t; \delta_c)$ between them and optimize the control point $\delta_c$, and (iii)

Table 1: **Success rates of adversarial attacks.** Results are reported on CIFAR-10 and ImageNet across three settings with 25 data cases each (single images in Setting A and image pairs in Settings B and C). "*ASR1*" and "*ASR2*" denote the success rate at each endpoint, "*ASR Both*" denotes the fraction of intermediate path points that successfully attack both endpoints simultaneously (in Setting A the endpoints coincide so ASR Both equals the path success), and "*ASR Avg*" denotes the average of ASR1 and ASR2 when defined. Attacks are evaluated along Bézier curves connecting the endpoints. All values are reported as mean ± standard deviation.

| Setting | Norm | CIFAR-10 | | | | ImageNet | | | |
|---|---|---|---|---|---|---|---|---|---|
| | | ASR1 | ASR2 | ASR Both | ASR Avg | ASR1 | ASR2 | ASR Both | ASR Avg |
| A | $\ell_\infty$ | N/A | N/A | 100.0±0.0 | 100.0±0.0 | N/A | N/A | 100.0±0.0 | 100.0±0.0 |
| | $\ell_2$ | N/A | N/A | 100.0±0.0 | 100.0±0.0 | N/A | N/A | 100.0±0.0 | 100.0±0.0 |
| | $\ell_1$ | N/A | N/A | 99.9±0.4 | 99.9±0.4 | N/A | N/A | 100.0±0.0 | 100.0±0.0 |
| B | $\ell_\infty$ | 98.6±1.6 | 98.3±1.8 | 97.0±2.4 | 98.5±1.2 | 99.5±0.9 | 99.4±0.9 | 98.9±1.4 | 99.4±0.7 |
| | $\ell_2$ | 97.8±1.7 | 97.7±1.8 | 95.4±2.9 | 97.7±1.4 | 99.3±1.1 | 99.3±1.1 | 98.6±1.9 | 99.3±1.0 |
| | $\ell_1$ | 87.2±32.0 | 75.0±42.2 | 62.3±46.6 | 81.1±23.4 | 100.0±0.0 | 99.7±0.7 | 99.7±0.7 | 99.8±0.4 |
| C | $\ell_\infty$ | 98.0±2.0 | 98.0±2.3 | 96.0±3.3 | 98.0±1.7 | 99.5±0.9 | 99.5±0.9 | 99.0±1.0 | 99.5±0.5 |
| | $\ell_2$ | 97.5±1.9 | 97.4±2.2 | 94.9±3.2 | 97.4±1.7 | 99.4±1.1 | 99.2±1.0 | 98.6±1.4 | 99.3±0.7 |
| | $\ell_1$ | 87.4±31.8 | 90.5±26.4 | 77.9±38.4 | 89.0±19.2 | 99.9±0.4 | 99.9±0.4 | 99.8±0.5 | 99.9±0.3 |

evaluate success by sampling $t \in [0.02, 0.98]$ and checking whether $f_\theta\big(x + \Pi_{\|\cdot\|_p \leq \epsilon}[B(t; \delta_c)]\big)$ is misclassified. Table 1 shows that optimizing the Bézier control point yields smooth adversarial paths that retain attack strength across intermediate points, supporting Bézier-based path construction and downstream uses (*e.g.*, Bézier crossover in MoCo-EA).

Table 2: **Transferability on CIFAR-10 and ImageNet.** "*Endp. Avg*" denotes the average success rate of the two endpoints, "*Path Succ.*" denotes the fraction of test images successfully attacked by at least one sampled point along the Bézier path, "*Imgs Resc.*" denotes the fraction of test images not attacked by endpoints but rescued by at least one path point, and "*Avg Pts*" denotes the average number of successful path points per image, with 50 points sampled per curve. All values are reported as mean ± standard deviation.

| Setting | Norm | CIFAR-10 | | | | ImageNet | | | |
|---|---|---|---|---|---|---|---|---|---|
| | | Endp. Avg | Path Succ. | Imgs Resc. | Avg Pts | Endp. Avg | Path Succ. | Imgs Resc. | Avg Pts |
| A | $\ell_\infty$ | $20.3 \pm 5.2$ | $34.7 \pm 5.2$ | $8.6 \pm 2.6$ | $12.7 \pm 2.5$ | $1.0 \pm 2.0$ | $3.5 \pm 5.7$ | $1.5 \pm 3.6$ | $0.8 \pm 1.5$ |
| | $\ell_2$ | $6.5 \pm 1.5$ | $9.4 \pm 2.2$ | $1.2 \pm 0.9$ | $3.6 \pm 0.7$ | $0.2 \pm 1.1$ | $0.5 \pm 2.2$ | $0.0 \pm 0.0$ | $0.0 \pm 0.1$ |
| | $\ell_1$ | $4.5 \pm 1.5$ | $11.2 \pm 2.3$ | $4.7 \pm 2.0$ | $3.4 \pm 0.8$ | $0.2 \pm 1.1$ | $2.0 \pm 4.0$ | $1.5 \pm 3.6$ | $0.7 \pm 1.4$ |
| B | $\ell_\infty$ | $20.4 \pm 3.6$ | $39.7 \pm 5.1$ | $11.8 \pm 3.2$ | $14.6 \pm 2.6$ | $0.5 \pm 1.5$ | $3.0 \pm 4.6$ | $2.0 \pm 4.0$ | $0.1 \pm 0.2$ |
| | $\ell_2$ | $6.5 \pm 1.3$ | $10.9 \pm 2.4$ | $1.8 \pm 1.7$ | $3.6 \pm 0.9$ | $0.0 \pm 0.0$ | $1.0 \pm 3.0$ | $1.0 \pm 3.0$ | $0.2 \pm 0.7$ |
| | $\ell_1$ | $4.8 \pm 1.4$ | $12.0 \pm 2.2$ | $4.5 \pm 1.7$ | $3.6 \pm 0.7$ | $0.0 \pm 0.0$ | $1.0 \pm 3.0$ | $1.0 \pm 3.0$ | $0.3 \pm 0.9$ |
| C | $\ell_\infty$ | $22.0 \pm 2.8$ | $38.2 \pm 4.3$ | $9.0 \pm 2.6$ | $13.9 \pm 2.0$ | $0.6 \pm 1.1$ | $2.8 \pm 3.3$ | $1.7 \pm 2.4$ | $0.7 \pm 1.1$ |
| | $\ell_2$ | $5.6 \pm 1.0$ | $9.9 \pm 1.9$ | $1.8 \pm 1.0$ | $2.9 \pm 0.8$ | $0.2 \pm 0.8$ | $1.2 \pm 2.2$ | $0.8 \pm 1.8$ | $0.2 \pm 0.6$ |
| | $\ell_1$ | $2.6 \pm 1.3$ | $8.4 \pm 2.9$ | $4.2 \pm 2.3$ | $2.4 \pm 0.7$ | $0.0 \pm 0.0$ | $0.8 \pm 1.8$ | $0.8 \pm 1.8$ | $0.1 \pm 0.4$ |

**Transferability Analysis.** We further investigate whether adversarial connectivity improves transferability across different images and settings. Specifically, we compare endpoint-average transferability with connectivity-based path transferability across $\ell_\infty$, $\ell_2$, and $\ell_1$ norms. Evaluation is conducted on unseen images using curves optimized from training cases. On CIFAR-10, we use 25 training samples per setting. On ImageNet, we use 20 training samples for Settings A and C, and 10 training samples for Setting B. Table 2 shows that connectivity-based paths consistently improve transferability across all norms and settings. This demonstrates that adversarial mode connectivity enables more robust and transferable perturbations, significantly enhancing the effectiveness of attacks beyond isolated adversarial examples. We hypothesize that the observed transferability gains arise because intermediate points along optimized paths frequently outperform the endpoints in transfer, indicating that the curve traverses flatter, more universal regions of the loss landscape. This may explain the observed improvements in reliability and cross-instance generalization.

**Multi-image Augmentation.** We study how adding $N$ auxiliary images when optimizing the Bézier control point affects transferability, varying auxiliary images $N \in \{0, 5, 10, 15, 20, 25\}$ with five

repetitions. Table 3 shows that adding auxiliary images yields gains in path success and rescue rate across all settings. Multi-image augmentation both regularizes the curve optimization and discovers more universal adversarial patterns that transfer to unseen images.

**Convergence and Sampling Density Analysis.** For each auxiliary-image count, we evaluate how *coverage*, the percentage of test images that are successfully attacked by at least one sampled point along the Bézier path, changes as the number of epochs increases. We consider epochs $\{10, 20, 30, 40, 50\}$, each repeated 5 times. We also evaluate two sampling densities along each Bézier curve, using 50 or 100 sampled points on the curve, and under each density report *coverage per point*, defined as the average number of images that a single sampled point successfully attacks. Table 4 shows that for a fixed auxiliary setting, increasing the number of optimization epochs generally improves coverage, and larger auxiliary sets achieve higher final coverage while typically requiring more epochs for the gains to fully materialize. Using more sampled points on each Bézier curve (100 vs. 50) yields slightly higher measured coverage per point, and the effect is stronger when more auxiliary images are available.

Table 3: **Effect of multi-image augmentation on CIFAR-10 under** $\ell_\infty$. "*Endp. Avg*" is the average endpoint transferability. "*Path Succ.*" is success rate when any intermediate point on the Bézier path succeeds. "*Imp.*" is the difference between Path Succ. and Endp. Avg. "*Rescue Rate*" is the fraction of test images that failed at endpoints but succeeded along the path. "*Aux*" denotes the number of auxiliary images used. Values are mean ± standard deviation.

| Setting | Aux | Endp. Avg | Path Succ. | Imp. | Rescue Rate |
|---|---|---|---|---|---|
| A | 0 | $18.2 \pm 1.6$ | $32.4 \pm 4.8$ | +14.2 | $8.0 \pm 3.3$ |
| | 5 | $18.2 \pm 1.6$ | $44.4 \pm 3.6$ | +26.2 | $20.2 \pm 2.6$ |
| | 10 | $18.2 \pm 1.6$ | $50.0 \pm 2.7$ | +31.8 | $25.6 \pm 2.9$ |
| | 15 | $18.2 \pm 1.6$ | $56.4 \pm 2.6$ | +38.2 | $31.8 \pm 1.3$ |
| | 20 | $18.2 \pm 1.6$ | $57.4 \pm 4.5$ | +39.2 | $33.0 \pm 3.0$ |
| | 25 | $18.2 \pm 1.6$ | $58.2 \pm 5.2$ | +40.0 | $33.8 \pm 4.9$ |
| B | 0 | $22.2 \pm 3.7$ | $43.4 \pm 7.5$ | +21.2 | $13.8 \pm 5.2$ |
| | 5 | $22.2 \pm 3.7$ | $49.2 \pm 5.4$ | +27.0 | $19.8 \pm 2.5$ |
| | 10 | $22.2 \pm 3.7$ | $54.4 \pm 4.9$ | +32.2 | $24.6 \pm 2.7$ |
| | 15 | $22.2 \pm 3.7$ | $58.6 \pm 5.8$ | +36.4 | $28.8 \pm 4.8$ |
| | 20 | $22.2 \pm 3.7$ | $60.8 \pm 4.1$ | +38.6 | $31.0 \pm 3.5$ |
| | 25 | $22.2 \pm 3.7$ | $61.8 \pm 6.6$ | +39.6 | $32.0 \pm 5.2$ |
| C | 0 | $21.7 \pm 2.8$ | $41.4 \pm 3.8$ | +19.7 | $12.2 \pm 4.1$ |
| | 5 | $21.7 \pm 2.8$ | $43.2 \pm 5.2$ | +21.5 | $13.8 \pm 6.2$ |
| | 10 | $21.7 \pm 2.8$ | $46.6 \pm 3.8$ | +24.9 | $17.0 \pm 6.2$ |
| | 15 | $21.7 \pm 2.8$ | $44.6 \pm 0.8$ | +22.9 | $15.2 \pm 2.9$ |
| | 20 | $21.7 \pm 2.8$ | $44.8 \pm 1.6$ | +23.1 | $15.2 \pm 3.7$ |
| | 25 | $21.7 \pm 2.8$ | $51.8 \pm 1.9$ | +30.1 | $22.0 \pm 4.6$ |

Table 4: **Convergence and sampling density.** Results are reported on CIFAR-10 under $\ell_\infty$. (a) Convergence across training epochs with 100 sampled points per curve, reported as coverage. (b) Coverage under different sampling densities, reported as coverage per point. "*Aux*" denotes the number of auxiliary images used. All values are mean ± standard deviation.

(a) Convergence vs. epochs (100 points).

| Setting | Aux | 10 epochs | 20 epochs | 30 epochs | 40 epochs | 50 epochs |
|---|---|---|---|---|---|---|
| A | 0 | 29.8±5.1 | 32.6±4.6 | 33.6±5.7 | 34.4±4.9 | 35.2±4.1 |
| | 5 | 37.0±2.9 | 42.2±4.8 | 44.8±5.4 | 46.2±3.1 | 46.4±3.3 |
| | 10 | 37.2±3.1 | 45.2±4.8 | 48.8±2.9 | 51.6±2.4 | 53.2±3.4 |
| | 15 | 38.8±2.5 | 49.4±4.8 | 56.4±4.2 | 58.6±2.9 | 58.4±2.5 |
| | 20 | 41.0±2.6 | 52.6±3.9 | 57.0±2.3 | 58.8±3.0 | 60.2±2.6 |
| | 25 | 43.6±5.3 | 56.2±5.3 | 60.2±3.8 | 60.4±5.4 | 61.6±5.0 |
| B | 0 | 40.4±6.8 | 43.0±8.2 | 44.4±7.6 | 43.6±6.8 | 44.2±6.8 |
| | 5 | 46.0±5.3 | 48.4±6.3 | 50.4±7.1 | 50.2±6.6 | 51.0±6.7 |
| | 10 | 45.8±6.4 | 51.0±4.6 | 54.2±3.8 | 56.0±4.5 | 56.2±4.4 |
| | 15 | 47.6±6.5 | 55.8±4.8 | 57.8±4.7 | 59.8±4.7 | 60.6±3.4 |
| | 20 | 50.0±6.8 | 58.0±2.9 | 60.8±3.1 | 60.6±4.4 | 60.6±5.0 |
| | 25 | 49.8±7.1 | 57.0±7.1 | 60.0±6.5 | 60.4±5.2 | 61.2±5.6 |
| C | 0 | 37.2±4.8 | 39.2±3.7 | 40.2±3.8 | 40.8±4.1 | 40.6±3.5 |
| | 5 | 38.4±3.4 | 41.0±3.4 | 43.4±3.3 | 44.2±4.5 | 45.4±5.2 |
| | 10 | 39.0±1.4 | 44.4±4.2 | 45.0±2.3 | 46.2±4.9 | 46.8±3.3 |
| | 15 | 39.2±2.6 | 42.6±4.3 | 45.2±2.9 | 46.2±3.4 | 46.2±2.9 |
| | 20 | 39.6±2.9 | 45.0±3.3 | 46.4±2.2 | 46.0±2.3 | 46.0±3.0 |
| | 25 | 40.6±3.3 | 49.0±2.6 | 52.2±4.4 | 53.4±4.2 | 53.4±4.3 |

(b) Coverage under sampling densities.

| Setting | Aux | 50 points | 100 points |
|---|---|---|---|
| A | 0 | 26.0±2.4 | 26.2±2.5 |
| | 5 | 37.0±2.2 | 37.1±2.2 |
| | 10 | 44.5±3.9 | 44.7±4.0 |
| | 15 | 50.6±2.6 | 50.9±2.6 |
| | 20 | 51.5±3.8 | 51.8±3.7 |
| | 25 | 53.7±5.2 | 54.1±5.3 |
| B | 0 | 34.4±7.6 | 34.4±7.6 |
| | 5 | 41.0±6.7 | 41.2±6.7 |
| | 10 | 47.8±4.2 | 48.1±4.2 |
| | 15 | 52.2±3.3 | 52.5±3.4 |
| | 20 | 53.0±4.3 | 53.3±4.3 |
| | 25 | 53.9±5.0 | 54.2±5.1 |
| C | 0 | 29.7±4.1 | 29.8±4.1 |
| | 5 | 35.1±4.1 | 35.2±4.2 |
| | 10 | 36.7±5.2 | 36.8±5.3 |
| | 15 | 36.4±2.1 | 36.6±2.2 |
| | 20 | 36.9±4.5 | 37.0±4.6 |
| | 25 | 44.7±4.6 | 44.9±4.7 |

Table 5: **Comparison of MoCo-EA and Traditional EA.** Results on CIFAR-10 and ImageNet with a population size of 30. "*Success rate*" denotes the percentage of successful attacks, "*Avg. generations*" denote the mean number of generations over successful cases, "*Avg. queries*" denote the mean number of queries, "*Avg. time*" denotes the mean runtime in seconds, and "*Rel. Imp.*", Relative improvement, denotes the percentage reduction of MoCo-EA compared to the baseline. All values are reported as mean ± standard deviation.

| Norm | Metric | CIFAR-10 | | | ImageNet | | |
|---|---|---|---|---|---|---|---|
| | | Traditional | MoCo-EA | Rel. Imp. | Traditional | MoCo-EA | Rel. Imp. |
| $\ell_\infty$ | Succ. rate | 93.3 | **100.0** | +6.7pp | 83.3 | **100.0** | +16.7pp |
| | Avg. gen. | $367.9 \pm 233.2$ | $\mathbf{1.7 \pm 1.1}$ | ↓99.5% | $456.8 \pm 309.0$ | $\mathbf{1.0 \pm 0.0}$ | ↓99.8% |
| | Avg. queries | $12329 \pm 8247$ | $\mathbf{628 \pm 367}$ | ↓94.9% | $16446 \pm 10408$ | $\mathbf{375 \pm 0}$ | ↓97.7% |
| | Avg. time | $29.44 \pm 19.72$ | $\mathbf{6.08 \pm 3.73}$ | ↓79.3% | $95.14 \pm 60.22$ | $\mathbf{5.05 \pm 0.04}$ | ↓94.7% |
| $\ell_2$ | Succ. rate | 6.7 | **100.0** | +93.3pp | 13.3 | **100.0** | +86.7pp |
| | Avg. gen. | $25.0 \pm 19.0$ | $\mathbf{1.4 \pm 1.6}$ | ↓94.4% | $24.8 \pm 9.8$ | $\mathbf{1.0 \pm 0.0}$ | ↓96.0% |
| | Avg. queries | $28052 \pm 7290$ | $\mathbf{513 \pm 561}$ | ↓98.2% | $26103 \pm 9936$ | $\mathbf{375 \pm 0}$ | ↓98.6% |
| | Avg. time | $67.94 \pm 17.67$ | $\mathbf{4.97 \pm 5.65}$ | ↓92.7% | $152.15 \pm 58.07$ | $\mathbf{5.01 \pm 0.02}$ | ↓96.7% |
| $\ell_1$ | Succ. rate | 56.7 | **100.0** | +43.3pp | 33.3 | **100.0** | +66.7pp |
| | Avg. gen. | $55.8 \pm 196.9$ | $\mathbf{1.0 \pm 0.5}$ | ↓98.2% | $13.3 \pm 22.6$ | $\mathbf{0.9 \pm 0.3}$ | ↓93.2% |
| | Avg. queries | $13966 \pm 14709$ | $\mathbf{375 \pm 178}$ | ↓97.3% | $20143 \pm 13945$ | $\mathbf{340 \pm 104}$ | ↓98.3% |
| | Avg. time | $34.82 \pm 36.64$ | $\mathbf{3.74 \pm 1.87}$ | ↓89.3% | $118.19 \pm 81.81$ | $\mathbf{4.61 \pm 1.48}$ | ↓96.1% |

## 5.3 MODE CONNECTIVITY EVOLUTIONARY ATTACK (MOCO-EA)

We first evaluate MoCo-EA method against the traditional evolutionary algorithm baseline, focusing on key performance outcomes. The baseline follows a standard population-based pipeline, and MoCo-EA keeps this pipeline unchanged, differing only in the crossover step, where it replaces element-wise crossover with our geometry-aware Bézier crossover. Additional details are provided in A.4. This comparison allows us to assess how the connectivity and transferability advantages of Bézier paths translate into practical improvements for evolutionary adversarial attacks.

Table 5 summarizes the comparative performance of MoCo-EA and the traditional evolutionary algorithm baseline on CIFAR-10 and ImageNet under the $\ell_\infty$, $\ell_2$, and $\ell_1$ perturbation norms. It is evident that MoCo-EA consistently surpasses the traditional EA across every performance dimension. It achieves near-perfect *success rates*, even under norm $\ell_2$ and $\ell_1$ constraints where the baseline often fails, while requiring only a handful of *generations* compared to the hundreds typically needed by the baseline. This efficiency translates into dramatically fewer *queries*, as offspring are sampled along optimized low-loss paths rather than through costly trial-and-error exploration. Consequently, the *runtime* improvements follow naturally, with MoCo-EA completing attacks substantially faster than its counterpart. Together, these results confirm that incorporating Bézier connectivity into evolutionary search yields uniformly superior performance by transforming recombination from random mixing into geometry-aware exploration of connected adversarial manifolds. Additionally, according to the ablation study in the appendix, population size primarily influences the efficiency of MoCo-EA, while its reliability remains consistently high across different population settings. A more detailed analysis can be viewed in A.6.

**MoCo-EA vs. Gradient-Based Attacks.** To Compare MoCo-EA with gradient-based adversarial attacks, we consider two experimental settings. First, we evaluate attack success rates on robustly trained models using the adversarially trained CIFAR-10 ResNet-50 checkpoint released in (Engstrom et al., 2019). We report *attack success rates* (ASR), defined as $100\%$ minus robust accuracy. All experiments are conducted on 100 randomly sampled test images using PGD (Madry et al., 2018), MI-FGSM (Dong et al., 2018), AutoAttack (AA) (Croce & Hein, 2020), Adaptive AutoAttack (AAA) (Liu et al., 2022), and our MoCo-EA. As shown in Table 6, MoCo-EA achieves higher success rates than the gradient-based baselines. Second, we evaluate performance under obfuscated gradient settings (Athalye et al., 2018) using a standard model on ImageNet. Specifically, we apply an additional quantization step to the input image, `torch.round(x × 5)/5`, which makes gradients vanish in most regions and therefore breaks conventional gradient-based attacks. All experiments are conducted on 100 randomly sampled test images. In these scenarios, gradient-based

attacks often become unreliable or even ineffective. In contrast, evolutionary algorithms rely partially on loss evaluations and remain fully applicable. As shown in Table 6, MoCo-EA consistently outperforms gradient-based methods under these challenging conditions. MoCo-EA is not intended to replace gradient-based attacks, but rather to highlight its distinct role in understanding adversarial geometry and in handling cases where gradients are unreliable or insufficient. Unlike conventional methods that follow a single optimization trajectory, MoCo-EA evolves an entire population of perturbations, enabling broader exploration of the loss landscape and reducing susceptibility to mode collapse. This population-based search is particularly advantageous in the two settings evaluated above, robust models and obfuscated gradient scenarios, where MoCo-EA consistently remains effective while traditional gradient-based attacks often struggle.

## 6 CONCLUSION

In this work, we introduced MoCo-EA, a novel approach that rethinks crossover operations in evolutionary adversarial attacks through continuous path optimization. By exploiting the mode connectivity property of adversarial perturbations, we demonstrated that successful attacks lie not on isolated points but on connected manifolds that can be traversed via optimized Bézier curves. Our experiments revealed that intermediate points along these paths exhibit superior transferability compared to endpoints,

Table 6: **Attack success rates (ASR, %) in two settings:** (a) robustly trained CIFAR-10 models and (b) obfuscated gradient defenses on ImageNet. Both evaluated on 100 randomly sampled test images under the $\ell_\infty$ norm.

| Setting | PGD | MIFGSM | AA | AAA | MoCo-EA |
|---|---|---|---|---|---|
| Robust model | 45 | 45 | 46 | 45 | **48** |
| Obfuscated gradients | 17 | 17 | 17 | 16 | **32** |

while replacing discrete genetic crossover with continuous Bézier interpolation yields significant improvements in both efficiency and effectiveness, achieving universal success across perturbation norms and reducing computational requirements by orders of magnitude. Beyond immediate benefits for adversarial attack generation, our findings highlight broader implications for understanding the geometric structure of adversarial space and suggest that defenses must consider the continuous nature of adversarial manifolds. Future work could investigate higher-order Bézier curves for more complex path optimization, explore defensive applications of adversarial connectivity, and extend our approach to other domains where evolutionary algorithms are applied.

REPRODUCIBILITY STATEMENT

The main paper provides detailed descriptions of datasets, preprocessing steps, model architectures, hyperparameters, training protocols, and evaluation metrics sufficient for replication. All experiments use publicly available datasets and standard splits where applicable. The complete code and scripts are included as supplementary materials accompanying this submission to enable independent verification.

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

# A  APPENDIX

## A.1  LARGE LANGUAGE MODEL (LLM) USAGE STATEMENT

In line with the ICLR 2026 guidelines, we disclose that large language models (LLMs) are used in two limited ways: (1) to improve the clarity and grammar of the writing, and (2) to assist retrieval and discovery of related work. All technical ideas, model designs, proofs, and experiments are conceived, implemented, and verified by the authors. Text suggested by LLMs is reviewed and edited for accuracy. LLMs are not used to generate research results or novel claims, and no sensitive or proprietary data are provided to external services.

## A.2  GEOMETRIC INTERPRETATION OF ADVERSARIAL MODE CONNECTIVITY IN ReLU NETWORKS.

Rather than claiming any formal proof, we provide an intuition for why adversarial perturbations produced by different optimization runs might nonetheless be joined by a smooth path that preserves adversarial behavior.

This intuition relies on the geometric structure arising from the particular ReLU activations and on qualitative observations regarding how misclassification regions extend across multiple linear regions.

A ReLU unit imposes a linear constraint of the form $w^\top x + b > 0$ or $w^\top x + b \leq 0$, and each such constraint defines a half-space. A fixed activation pattern corresponds to the finite intersection of such half-spaces, forming a convex polytope. Inside such a region the network is affine, $f(x) = Ax + c$, and the decision boundary is an affine hyperplane. Therefore, in the special case where two adversarial perturbations fall inside the same activation region, connectivity is immediate: convexity ensures that the straight-line interpolation between the two points remains in the region, and so does any Bézier curve whose control points lie within that region.

In realistic settings, however, adversarial perturbations found by PGD from different initializations typically lie in different activation regions. In this more general case, connectivity cannot be explained by the geometry of a single polytope. Instead, the relevant structure is the misclassification region:

$$\mathcal{A} = \{\, \delta \ \mid \ f(x + \delta) \neq y, \ \|\delta\|_p \leq \epsilon \,\},$$

which is the union of the misclassified subsets of many activation regions. Although we make no formal topological claim, several empirical regularities of piecewise-linear networks suggest that $\mathcal{A}$ tends to form a large connected set in practice.

First, because the affine classifiers associated with neighboring activation regions are pieces of the same global piecewise-linear function, their decision hyperplanes usually meet continuously along the shared facets of adjacent regions. As a result, misclassified portions of these regions often touch or overlap along those facets rather than breaking into isolated fragments.

Second, adversarial perturbations often move along shallow or low-curvature directions of the loss surface. Such directions extend across many activation regions and vary smoothly as the activation pattern changes. This produces broad "corridors" of misclassification that traverse multiple linear regions, making it more likely that $\delta_1$ and $\delta_2$ obtained from different PGD runs lie in the same connected misclassification component, even when their activation patterns differ.

Under this structure, a Bézier curve may cross activation-region boundaries while remaining inside the misclassification region. By optimizing the control point $\delta_c$ to maximize adversarial loss along the curve, the curve is bent toward shared low-loss directions of the misclassification set. This procedure steers the path away from regions of correct classification and toward low-loss passages that span multiple activation regions.

Empirically, the optimized curve remains adversarial for almost all sampled values of $t$, indicating that adversarial perturbations belong not to isolated pockets but to a large connected adversarial basin that extends across many activation regions. This geometric viewpoint also explains why intermediate points along the optimized Bézier curve often generalize better: compared to the PGD

endpoints, which correspond to sharper local optima tied to particular activation patterns, the intermediate points lie deeper inside the shared misclassification component where the loss surface is flatter and adversarial properties are more stable.

### A.3 MoCo-EA Algorithm

Complete pseudocode for the Mode Connectivity Evolutionary Attack (MoCo-EA) is provided in Algorithm 2.

**InitializePopulation** Given an input image $x$, a budget $\epsilon$, and population size $N$, this routine samples $N$ initial perturbations $\{\delta_i\}_{i=1}^N$ either randomly or using PGD, inside the feasible $\ell_p$-ball, i.e., $\|\delta_i\|_p \leq \epsilon$, and returns the set $P$.

**EvaluateFitness** For each candidate $\delta \in P$, the fitness is computed from an attack objective such as the negative classification loss or a success indicator on $f_\theta(x + \delta)$ against label $y$. Typical choices include $-\mathcal{L}(f_\theta(x + \delta), y)$ for untargeted attacks or the margin/targeted loss. The routine returns a vector $F$ of fitness scores aligned with $P$.

**elite $\cup$ SelectBest** Elitism first preserves the top-$k$ individuals from the current population $P$ according to $F$ as the set $\texttt{elite} = \texttt{SelectTop}(P, k)$. Then, among the newly generated offspring, $\texttt{SelectBest}(\cdot, N - k)$ picks the highest-fitness $(N - k)$ candidates. Their union $\texttt{elite} \cup \texttt{SelectBest}(\cdot)$ forms the next generation of size $N$ while guaranteeing that the strongest current solutions are never discarded.

---

**Algorithm 2** Mode Connectivity Evolutionary Attack (MoCo-EA)

---

1: **Input:** Image $x$, label $y$, model $f_\theta$, max generations $G$
2: **Parameters:** population size $N$, elite size $k$, mutation rate $p_m$, mutation std $\sigma$
3: **Output:** Adversarial perturbation $\delta^*$
4: $P \leftarrow$ InitializePopulation$(N, \epsilon)$
5: $\delta^* \leftarrow$ null
6: **for** $g = 1$ **to** $G$ **do**
7:      $F \leftarrow$ EvaluateFitness$(P, x, y, f_\theta)$
8:      **if** $\max(F) >$ fitness$(\delta^*)$ **then**
9:          $\delta^* \leftarrow \arg\max_{\delta \in P}$ fitness$(\delta)$
10:      **end if**
11:      **if** IsSuccessful$(\delta^*)$ **then**
12:          **return** $\delta^*$
13:      **end if**
14:      $parents \leftarrow$ TournamentSelection$(P, F)$
15:      $offspring \leftarrow \varnothing$
16:      **for each** $(p_1, p_2)$ **in** $parents$ **do**
17:          $(c_1, c_2) \leftarrow$ BezierCrossover$(p_1, p_2, x, y, f_\theta)$
18:          $offspring \leftarrow offspring \cup \{$Mutate$(c_1, p_m, \sigma)$, Mutate$(c_2, p_m, \sigma)\}$
19:      **end for**
20:      $elite \leftarrow$ SelectTop$(P, k)$
21:      $P \leftarrow elite \cup$ SelectBest$(offspring, N - k)$
22: **end for**
23: **return** $\delta^*$

---

**Remark.** The crossover operator used here is the geometry-aware Bézier subroutine (Algorithm 1 in the main text). See Section 4.2 for the algorithmic overview and Algorithm 1 for implementation details of the crossover.

### A.4 Experimental Setup Details

**CIFAR-10 and ResNet-18 details.** CIFAR-10 contains 50,000 training and 10,000 test images across 10 classes (Krizhevsky, 2009). We use a ResNet-18 (He et al., 2016) adapted for CIFAR-10 by replacing the initial $7 \times 7$ convolution with a $3 \times 3$ kernel (stride $= 1$, padding $= 1$) and removing the max-pooling layer. The model is trained for 200 epochs using SGD with momentum 0.9, weight

decay $5 \times 10^{-4}$, and a multi-step learning-rate schedule (initial lr $= 0.1$, decayed by $10\times$ at epochs $60, 120, 160$), achieving $95.1\%$ clean test accuracy.

**ImageNet and ViT-Base/16 details.** For ImageNet (Deng et al., 2009), we evaluate on the standard validation set (50,000 images, 1,000 classes). We adopt a Vision Transformer (ViT-Base, patch size $16\times16$) (Dosovitskiy et al., 2021) pretrained on ImageNet. Preprocessing follows the common pipeline: resize the shorter side to 256, center crop to $224\times224$, and normalize with the pretrained ViT statistics (mean $= 0.5$, std $= 0.5$). The pretrained ViT achieves $84.4\%$ top-1 accuracy on the validation set.

**Image selection protocol for Settings A/B/C.** The connectivity scenarios are fixed across datasets. On **CIFAR-10**: Setting A uses a single image from class *cat*; Setting B uses two *cat* images from the same class; Setting C pairs a *cat* image with a *dog* image. On **ImageNet**: the same structure is applied with *Egyptian cat* images for Settings A and B, and with an *Egyptian cat* image paired with a *Labrador retriever* image for Setting C.

**Details of the Baseline Evolutionary Algorithm.** The baseline evolutionary attack follows a standard population-based procedure (Alzantot et al., 2019). It maintains a population of $N$ perturbations, where 30 is the default, and iteratively evolves them under the same $\ell_p$-norm ball. Each perturbation is initialized uniformly from the $\ell_p$ ball of radius $\epsilon$. For each candidate $\boldsymbol{\delta}$, fitness is evaluated as the negative cross-entropy loss of $f_\theta(x+\delta)$, which directly reflects adversarial strength. At every iteration, we preserve the top $K = 5$ elite individuals, while the remaining candidates for reproduction are selected via tournament selection with size $k = 3$. New offspring are generated using uniform crossover with probability $p = 0.5$, where each pixel is independently inherited from one of the two parents. After crossover, Gaussian mutation with standard deviation $0.02\epsilon$ is applied with probability 0.2, and the resulting perturbations are projected back onto the $\ell_p$ ball to satisfy the norm constraint. The next generation is formed by combining the preserved elites with the highest-fitness offspring. This iterative process is repeated for a maximum of $T = 1000$ iterations or until the model misclassifies the image.

## A.5 Additional Transferability Results

Additional transferability results that complement the main paper. The tables compile CIFAR-10 and ImageNet evaluations under Settings A/B/C and norms ($\ell_\infty, \ell_2, \ell_1$).

Table 7: **Transferability on CIFAR-10.** Results are reported on an additional set of 100 test images. Bézier curves are optimized on 25 training cases (Setting A uses single-image endpoints, Settings B and C use image pairs). "*Endp. Avg*" denotes the average success rate of the two endpoints, "*Path Succ.*" the percentage of test images successfully attacked by at least one sampled point along the Bézier path, "*Imgs Resc.*" the fraction of images not attacked by endpoints but rescued by at least one path point, "*Avg Pts/Img*" the average number of successful path points per image (50 sampled per curve), and "*Improv.*" the improvement of Path Succ. over Endp. Avg. All values are reported as mean ± standard deviation.

| Setting | Norm | Endp. Avg | Path Succ. | Imgs Resc. | Avg Pts/Img | Improv. |
|---------|------|-----------|------------|------------|-------------|---------|
| A | $\ell_\infty$ | $20.3 \pm 5.2$ | $34.7 \pm 5.2$ | $8.6 \pm 2.6$ | $12.7 \pm 2.5/50$ | $+14.4$ |
|   | $\ell_2$ | $6.5 \pm 1.5$ | $9.4 \pm 2.2$ | $1.2 \pm 0.9$ | $3.6 \pm 0.7/50$ | $+2.9$ |
|   | $\ell_1$ | $4.5 \pm 1.5$ | $11.2 \pm 2.3$ | $4.7 \pm 2.0$ | $3.4 \pm 0.8/50$ | $+6.7$ |
| B | $\ell_\infty$ | $20.4 \pm 3.6$ | $39.7 \pm 5.1$ | $11.8 \pm 3.2$ | $14.6 \pm 2.6/50$ | $+19.3$ |
|   | $\ell_2$ | $6.5 \pm 1.3$ | $10.9 \pm 2.4$ | $1.8 \pm 1.7$ | $3.6 \pm 0.9/50$ | $+4.4$ |
|   | $\ell_1$ | $4.8 \pm 1.4$ | $12.0 \pm 2.2$ | $4.5 \pm 1.7$ | $3.6 \pm 0.7/50$ | $+7.2$ |
| C | $\ell_\infty$ | $22.0 \pm 2.8$ | $38.2 \pm 4.3$ | $9.0 \pm 2.6$ | $13.9 \pm 2.0/50$ | $+16.3$ |
|   | $\ell_2$ | $5.6 \pm 1.0$ | $9.9 \pm 1.9$ | $1.8 \pm 1.0$ | $2.9 \pm 0.8/50$ | $+4.4$ |
|   | $\ell_1$ | $2.6 \pm 1.3$ | $8.4 \pm 2.9$ | $4.2 \pm 2.3$ | $2.4 \pm 0.7/50$ | $+5.8$ |

Table 8: **Transferability on ImageNet.** Results are reported with a fixed test set of 10 images and deterministic training samples (20 for Setting A and C, 10 for Setting B). "*Endp. Avg*" denotes the average success rate of the two endpoints $(\delta_1, \delta_2)$, "*Path Succ.*" denotes the percentage of test images successfully attacked by at least one point along the Bézier path, "*Imgs Resc.*" denotes the fraction of test images not attacked by endpoints but rescued by at least one path point, "*Avg Pts/Img*" denotes the average number of successful path points per image (out of 50), and "*Improv.*" denotes the improvement of Path Succ. over Endp. Avg. Attacks use $\ell_\infty$ with $\epsilon = 8/255$, $\ell_2$ with $\epsilon = 4.0$, and $\ell_1$ with $\epsilon = 300.0$ to produce usable PGD endpoints. All values are reported as mean $\pm$ standard deviation.

| Setting | Norm | Endp. Avg | Path Succ. | Imgs Resc. | Avg Pts/Img | Improv. |
|---------|------|-----------|------------|------------|-------------|---------|
| A | $\ell_\infty$ | $1.0 \pm 2.0$ | $3.5 \pm 5.7$ | $1.5 \pm 3.6$ | $0.8 \pm 1.5/50$ | $+2.5$ |
|   | $\ell_2$ | $0.2 \pm 1.1$ | $0.5 \pm 2.2$ | $0.0 \pm 0.0$ | $0.0 \pm 0.1/50$ | $+0.2$ |
|   | $\ell_1$ | $0.2 \pm 1.1$ | $2.0 \pm 4.0$ | $1.5 \pm 3.6$ | $0.7 \pm 1.4/50$ | $+1.8$ |
| B | $\ell_\infty$ | $0.5 \pm 1.5$ | $3.0 \pm 4.6$ | $2.0 \pm 4.0$ | $0.1 \pm 0.2/50$ | $+2.5$ |
|   | $\ell_2$ | $0.0 \pm 0.0$ | $1.0 \pm 3.0$ | $1.0 \pm 3.0$ | $0.2 \pm 0.7/50$ | $+1.0$ |
|   | $\ell_1$ | $0.0 \pm 0.0$ | $1.0 \pm 3.0$ | $1.0 \pm 3.0$ | $0.3 \pm 0.9/50$ | $+1.0$ |
| C | $\ell_\infty$ | $0.6 \pm 1.1$ | $2.8 \pm 3.3$ | $1.7 \pm 2.4$ | $0.7 \pm 1.1/50$ | $+2.1$ |
|   | $\ell_2$ | $0.2 \pm 0.8$ | $1.2 \pm 2.2$ | $0.8 \pm 1.8$ | $0.2 \pm 0.6/50$ | $+1.0$ |
|   | $\ell_1$ | $0.0 \pm 0.0$ | $0.8 \pm 1.8$ | $0.8 \pm 1.8$ | $0.1 \pm 0.4/50$ | $+0.8$ |

### A.6 MORE ANALYSIS OF MoCo-EA AND ITS ABLATION STUDY

To supplement the main text, we provide a more detailed examination of Table 5, examining each performance metric (success rate, generations, queries, and runtime) and explaining the factors behind MoCo-EA's improvements.

*Success Rate:* MoCo-EA achieves consistently higher success rates than the traditional EA across all datasets and perturbation norms. While the baseline often struggles under less restrictive settings (particularly under $\ell_2$ and $\ell_1$ constraints), MoCo-EA attains near-perfect attack success across all tested settings. This improvement highlights the practical benefits of integrating Bézier connectivity: by ensuring that offspring perturbations lie on low-loss paths between adversarial modes, MoCo-EA preserves adversarial validity throughout the evolutionary process. The consistently superior success rates therefore demonstrate that connectivity and transferability properties observed in earlier analyses directly translate into more reliable attack generation within the evolutionary framework.

*Average Generations:* MoCo-EA converges within only a few generations, in stark contrast to the baseline EA which often requires hundreds of iterations to identify effective adversarial perturbations. This sharp reduction in generational cost illustrates the efficiency of the Bézier crossover operator: instead of relying on random recombination that frequently disrupts adversarial structure, offspring are sampled along optimized low-loss trajectories that reliably preserve attack validity. As a result, MoCo-EA transforms the evolutionary process from a slow, trial-and-error search into a rapid and directed exploration of adversarial space.

*Average Queries:* MoCo-EA requires dramatically fewer model queries compared to the traditional EA baseline. By exploiting Bézier connectivity, the search process is guided toward regions of perturbation space that are already adversarially valid, thereby reducing the need for extensive query-based exploration. This efficiency gain is particularly significant under every norm constraints, where traditional EA must rely on large query budgets to locate viable perturbations. The reduction in query complexity underscores the practical value of geometry-aware crossover, making MoCo-EA more applicable in realistic scenarios where query access to the target model is limited or costly.

*Average Runtime:* The improvements in average runtime follow naturally from the substantial reductions in generations and queries. Since MoCo-EA converges quickly and requires far fewer interactions with the target model, its wall-clock time is consistently lower than that of the traditional EA baseline. This outcome is therefore an expected consequence of the algorithm's efficiency gains, further confirming the practicality of integrating Bézier connectivity into evolutionary attacks.

**Effect of Population Size.** We further analyze the effect of population size (15/30/45) under $\ell_\infty$, $\ell_2$, and $\ell_1$ norms on CIFAR-10, as summarized in Table 9. Compared to Table 5, which reports results

Table 9: **Effect of varying population size under $\ell_\infty$, $\ell_2$, and $\ell_1$ on CIFAR-10.** "*SR*" denotes success rate (%), "*Gen*" the average number of generations to success (successful cases only), "*Query*" the average number of queries, and "*Time*" the average runtime in seconds. Results are reported for population sizes 15, 30, and 45, with each cell showing Traditional / MoCo-EA.

| Norm | Metric | 15 | 30 | 45 |
|------|--------|----|----|----|
| | | Traditional / MoCo-EA | Traditional / MoCo-EA | Traditional / MoCo-EA |
| $\ell_\infty$ | SR (%) | 76.7 / **100.0** | 93.3 / **100.0** | 96.7 / **100.0** |
| | Gen | $487.8\pm221.2$ / $\mathbf{1.9\pm1.3}$ | $367.9\pm233.2$ / $\mathbf{1.7\pm1.1}$ | $315.9\pm231.0$ / $\mathbf{1.7\pm0.9}$ |
| | Query | $9122\pm4354$ / $\mathbf{317\pm208}$ | $12329\pm8247$ / $\mathbf{628\pm367}$ | $15286\pm11614$ / $\mathbf{890\pm460}$ |
| | Time (s) | $21.63\pm10.24$ / $\mathbf{2.96\pm1.78}$ | $29.44\pm19.72$ / $\mathbf{6.08\pm3.73}$ | $36.36\pm27.60$ / $\mathbf{8.44\pm4.53}$ |
| $\ell_2$ | SR (%) | 3.3 / **100.0** | 6.7 / **100.0** | 10.0 / **100.0** |
| | Gen | $7.0\pm0.0$ / $\mathbf{2.4\pm6.6}$ | $25.0\pm19.0$ / $\mathbf{1.4\pm1.6}$ | $20.7\pm11.1$ / $\mathbf{1.7\pm3.4}$ |
| | Query | $14504\pm2671$ / $\mathbf{398\pm1074}$ | $28052\pm7290$ / $\mathbf{513\pm561}$ | $40598\pm13208$ / $\mathbf{907\pm1728}$ |
| | Time (s) | $35.05\pm6.46$ / $\mathbf{3.91\pm10.77}$ | $67.94\pm17.67$ / $\mathbf{4.97\pm5.65}$ | $97.78\pm31.80$ / $\mathbf{8.88\pm17.58}$ |
| $\ell_1$ | SR (%) | 40.0 / **100.0** | 56.7 / **100.0** | 70.0 / **100.0** |
| | Gen | $96.8\pm170.0$ / $\mathbf{1.0\pm0.5}$ | $55.8\pm196.9$ / $\mathbf{1.0\pm0.5}$ | $8.2\pm7.3$ / $\mathbf{1.0\pm0.4}$ |
| | Query | $9587\pm6823$ / $\mathbf{177\pm84}$ | $13966\pm14709$ / $\mathbf{375\pm178}$ | $13790\pm20434$ / $\mathbf{535\pm206}$ |
| | Time (s) | $23.95\pm17.03$ / $\mathbf{1.75\pm0.88}$ | $34.82\pm36.64$ / $\mathbf{3.74\pm1.87}$ | $34.52\pm51.10$ / $\mathbf{5.31\pm2.18}$ |

at a fixed population size, this ablation reveals how varying the population influences performance across the three norms. Three observations emerge:

*(i) Success rate vs. population size.* For the traditional EA, increasing the population improves success rate but leaves it far from reliable under $\ell_2$ (*e.g.*, $3.3\% \rightarrow 10.0\%$ as population grows from 15 to 45), and still below 100% under $\ell_\infty/\ell_1$. In contrast, MoCo-EA attains 100% success across all tested populations and norms, indicating that geometry-aware crossover removes the method's reliance on large populations to achieve reliability.

*(ii) Generational cost and its interpretability.* For the traditional EA, average generations decrease as population grows under $\ell_\infty$ and $\ell_1$ (*e.g.*, $487.8 \rightarrow 315.9$ and $96.8 \rightarrow 8.2$), consistent with diversity aiding convergence. However, under $\ell_2$ the trend is inconsistent ($7.0 \rightarrow 25.0 \rightarrow 20.7$). This inconsistency is rational, because the metric is computed only over successful attacks: when success is rare, the estimate becomes sample-size sensitive and is not representative of the algorithm's typical behavior. MoCo-EA, by contrast, converges in about one to two generations across all populations and norms, with small dispersion, reflecting a search guided along adversarially valid low-loss paths.

*(iii) Query/runtime scaling with population.* For the traditional EA, average queries and wall-clock time *increase* with population across all norms (*e.g.*, $\ell_\infty$ queries $9122 \rightarrow 15286$; $\ell_2$ time $35.05\text{s} \rightarrow 97.78\text{s}$), because per-generation evaluation cost grows with the number of individuals and the generational reduction is insufficient to offset this. MoCo-EA exhibits the same *linear-like* scaling in queries/time with population (*e.g.*, $\ell_\infty$ queries $317 \rightarrow 890$), but since it typically converges in one or two generations, the absolute cost remains low (single-digit seconds), and the success rate does not benefit from larger populations. Consequently, smaller populations (*e.g.*, 15) already deliver the desired reliability and minimize query/time budgets.

Table 5 demonstrates MoCo-EA's advantage at a population size of 30. The ablation in Table 9 further shows that this advantage is *robust* across population sizes: MoCo-EA maintains 100% success and near-constant generational cost for $15 \leq$ population $\leq 45$, while its query/time overhead grows approximately with population size. Traditional EA, in contrast, exhibits a classical exploration–efficiency trade-off: larger populations yield somewhat higher success rates and fewer generations under $\ell_\infty/\ell_1$, but at the price of substantially higher queries and time, and still fail to produce reliable success under $\ell_2$.

Table 10: **Statistical Tests.** Mean $\pm$ standard deviation of the traditional EA over 5 seeds and the corresponding one-sided paired t-test p-values comparing against MoCo-EA on ImageNet.

| Norm | Metric | Traditional EA) | p-value |
|------|--------|-----------------|---------|
| $\ell_\infty$ | Succ. rate | $87.3 \pm 2.8$ | $2.65 \times 10^{-4}$ |
| | Gen. | $456.8 \pm 309.0$ | $9.06 \times 10^{-8}$ |
| $\ell_2$ | Succ. rate | $12.0 \pm 1.8$ | $2.22 \times 10^{-8}$ |
| | Gen. | $24.8 \pm 9.8$ | $1.23 \times 10^{-2}$ |
| $\ell_1$ | Succ. rate | $35.3 \pm 4.5$ | $2.73 \times 10^{-6}$ |
| | Gen. | $13.3 \pm 22.6$ | $6.29 \times 10^{-2}$ |

Population size acts as a critical efficiency knob rather than a reliability enabler for MoCo-EA, because geometry-aware crossover already ensures connected, low-loss exploration, increasing the population provides no success-rate gain and only inflates queries/runtime. Hence, MoCo-EA's reliability is population-insensitive on CIFAR-10 across all tested norms, and its most resource-efficient regime is attained at smaller populations. For the traditional EA, larger populations partially compensate for unguided recombination by improving success and reducing generations under $\ell_\infty/\ell_1$, but they remain inefficient and ineffective under $\ell_2$, underscoring the central role of the geometry-aware prior introduced by Bézier connectivity.

**Statistical Tests.** We conducted one-sided paired t-tests on ImageNet results in Table 5 to evaluate whether the improvements of MoCo-EA over the traditional evolutionary attack are statistically significant. Table 10 show that for success rates, the p-values are all much smaller than $0.05$, which means the differences are statistically significant. For the number of generations, the p-values indicate statistically significant differences for $\ell_\infty$ and $\ell_2$ ($p < 0.05$), while $\ell_1$ still shows a clear improving trend in favor of MoCo-EA.

**Computational Complexity Analysis.** The primary efficiency gain of our method does not come from reducing the per-generation cost, but from its substantially faster convergence. Because Bézier crossover incorporates a mode-connectivity–guided mechanism, MoCo-EA reaches high-quality perturbations in far fewer generations. Empirically, as shown in Table 5, this yields a 94–99% reduction in total queries (i.e., the number of forward model evaluations), resulting in strictly lower overall runtime than the baseline EA.

Let $C_{fw}$ and $C_{bw}$ denote the cost of one forward and backward model evaluation, respectively, and let $C_{grad} = C_{fw} + C_{bw}$. Let $N$ be the population size, $d$ the input dimension, and $m$ the number of modified entries during crossover or mutation (with $m \ll d$ in high-dimensional inputs).

For baseline EA, fitness evaluation requires one forward pass per individual, giving a per-generation cost of $O(N \cdot C_{fw})$. Uniform crossover does not operate on all $d$ coordinates, in our implementation it modifies only $m$ selected entries, so its cost is $O(m)$ per offspring. Gaussian mutation and projection are also $O(m)$. Thus, the baseline EA per generation cost is: $O(N \cdot C_{fw} + N \cdot m)$

For MoCo-EA, Bézier crossover introduces a small backward component. Instead of operating on $d$-dimensional vectors, it optimizes a control point with $\tau$ gradient steps, each requiring one forward and backward pass. It then evaluates $k$ points on the curve, requiring $k$ forward passes. Therefore, the additional overhead per parent pair is: $O(\tau \cdot C_{grad} + k \cdot C_{fw})$. Since $\tau$ and $k$ are fixed small constants ($\tau = 5$, $k = 3$), the overall per-generation complexity of MoCo-EA remains: $O(N \cdot C_{fw})$ identical to the baseline EA up to a small constant factor contributed by the Bézier crossover.

The extra backward computations introduced by Bézier crossover are bounded by a fixed small constant and add only negligible per-query overhead compared to the dominant model-evaluation cost. The overall efficiency improvement comes from MoCo-EA's much faster convergence: the geometry-guided crossover requires far fewer generations and queries, yielding a substantially lower end-to-end runtime than the baseline EA.

