# OpenReview forum: "MoCo-EA: Exploiting Adversarial Mode Connectivity for Efficient Evolutionary Attacks"
_ICLR.cc/2026/Conference — ICLR 2026 Conference Desk Rejected Submission_

### Official Review · Reviewer_D1Af · 2025-10-26

**Soundness:** 3
**Presentation:** 3
**Contribution:** 3
**Rating:** 6
**Confidence:** 4

**Summary:**

This paper proposes the Mode Connectivity Evolutionary Attack (MoCo-EA), which replaces traditional crossover operations with a novel Bezier crossover operator to optimize perturbations along continuous Bezier curves. By revealing the geometric structure of adversarial space and leveraging principled path optimization, MoCo-EA transforms evolutionary attacks from slow and unreliable processes into efficient and dependable methods.

**Strengths:**

1、The paper correctly identifies a key limitation of gradient-based methods: their local operation around single examples fails to exploit the global structure that may connect different adversarial modes. This motivates the study of continuous connectivity between successful perturbations.

2、Evolutionary algorithms are rightly acknowledged for their effectiveness in black-box scenarios due to their ability to explore complex, non-differentiable search spaces.

3、Unlike prior methods that treat perturbations as isolated points, the introduction of Bezier-based crossover—enabling gradient-informed path interpolation between adversarial modes—represents a novel and structured approach. This not only preserves adversarial properties but also enhances sample diversity and transferability.

4、The experimental results (e.g., Table 5) convincingly demonstrate that MoCo-EA achieves near-perfect success rates with dramatically fewer generations, queries, and runtime compared to traditional evolutionary attacks, validating the practical benefits of the proposed method.

**Weaknesses:**

1、Lack of computational complexity analysis. The paper does not provide a formal analysis of the computational complexity of the Bezier crossover operation or the overall MoCo-EA algorithm. While empirical results show efficiency gains, a theoretical comparison of time/space complexity with traditional crossover operators would strengthen the contribution.

**Questions:**

1、Include a computational complexity analysis of Bezier crossover relative to traditional crossover, detailing both theoretical and practical overhead.

2、Strengthen the theoretical motivation or provide more formal insights into why adversarial mode connectivity holds and why intermediate points generalize better.

---

> ### Author Response · Authors · 2025-11-21
> **To Reviewer D1Af (1/2)**
>
> We thank the reviewer for highlighting the strengths of our work, including the motivation, novelty of the Bézier-based crossover, and the empirical gains of MoCo-EA. Regarding the noted weakness on computational complexity and the request for stronger theoretical motivation, we have added clarifications and analyses in the revised manuscript and address these points in detail below.
>
> ### **Q1) Include a computational complexity analysis of Bezier crossover relative to traditional crossover, detailing both theoretical and practical overhead.**
>
> The primary efficiency gain of our method does not come from reducing the per-generation cost, but from its substantially faster convergence. Because Bézier crossover incorporates a mode-connectivity–guided mechanism, MoCo-EA reaches high-quality perturbations in far fewer generations. Empirically, as shown in Table 5, this yields a 94–99\% reduction in total queries (i.e., the number of forward model evaluations), resulting in strictly lower overall runtime than the baseline EA.
> Let $C_{fw}$ and $C_{bw}$ denote the cost of one forward and backward model evaluation, respectively, and let $C_{grad} = C_{fw} + C_{bw}$. Let $N$ be the population size, $d$ the input dimension, and $m$ the number of modified entries during crossover or mutation (with $m \ll d$ in high-dimensional inputs).
>
> For baseline EA, fitness evaluation requires one forward pass per individual, giving a per-generation cost of $O(N \cdot C_{fw})$. Uniform crossover does not operate on all $d$ coordinates; in our implementation, it modifies only $m$ selected entries, so its cost is $O(m)$ per offspring. Gaussian mutation and projection are also $O(m)$. Thus, the baseline EA per generation cost is: $O(N \cdot C_{fw} + N \cdot m)$
>
> For MoCo-EA, Bézier crossover introduces a small backward component. Instead of operating on $d$-dimensional vectors, it optimizes a control point with $\tau$ gradient steps, each requiring one forward and backward pass. It then evaluates $k$ points on the curve, requiring $k$ forward passes. Therefore, the additional overhead per parent pair is: $O(\tau \cdot C_{grad} + k \cdot C_{fw})$. Since $\tau$ and $k$ are fixed small constants ($\tau$ = 5, $k$ = 3), the overall per-generation complexity of MoCo-EA remains: $O(N \cdot C_{fw})$ identical to the baseline EA up to a small constant factor contributed by the Bézier crossover.
>
> The extra backward computations introduced by Bézier crossover are bounded by a fixed small constant and add only negligible per-query overhead compared to the dominant model-evaluation cost. The overall efficiency improvement comes from MoCo-EA’s much faster convergence: the geometry-guided crossover requires far fewer generations and queries, yielding a substantially lower end-to-end runtime than the baseline EA. We include this full computational complexity analysis in Appendix A.6.

---

> ### Author Response · Authors · 2025-11-21
> **To Reviewer D1Af (2/2)**
>
> ### **Q2) Strengthen the theoretical motivation or provide more formal insights into why adversarial mode connectivity holds and why intermediate points generalize better.**
>
> We emphasize that our work is primarily an empirical study of adversarial mode connectivity. We provide additional intuition and motivation below.
>
> **1. Why does adversarial mode connectivity hold?**
>
> Classical mode-connectivity results show that two different trained models can often be connected by a smooth path in weight space along which the training loss stays low [1]. Our adversarial mode-connectivity result follows the same high-level principle, but applied to input perturbations rather than weights, and with the objective reversed: instead of trying to keep the loss low along the path (as classical mode connectivity does), we look for a path that keeps the loss high at every point so that all points remain adversarial. To do this, we consider a family of quadratic Bézier curves connecting two adversarial perturbations. Each curve is controlled by a single "control point" that determines the shape of the trajectory between the two endpoints. Because the set of possible control-point locations is bounded and continuous, and because the model's loss changes continuously with the input, there is always at least one control-point choice that maximizes the lowest loss value along the entire curve.
>
> Our Bézier optimization indeed discovers such paths consistently across norms, datasets, and connectivity settings, providing constructive empirical evidence that successful adversarial perturbations lie on connected high-loss manifolds rather than being isolated points.
>
> **2. Why do intermediate Bézier points generalize better?**
>
> PGD optimizes a single perturbation for one image and therefore tends to converge to sharp, highly localized adversarial maxima with poor transferability [2]. In contrast, Bézier optimization evaluates multiple sampled points along the curve simultaneously, so sharp peaks cannot satisfy this multi-point constraint. As a result, the trajectory is naturally pulled toward broader, flatter, and more stable high-loss regions.
>
> Evolutionary search further amplifies this effect: the two parents may originate from different adversarial basins, and the optimized Bézier curve must connect these basins while maintaining high loss at many sampled values. Consequently, intermediate points lie deeper inside flat and universal high-loss directions, which explains their consistently superior transferability.
>
> When auxiliary images are included, the control point must induce high loss on multiple inputs simultaneously. This acts as multi-task regularization and drives the path into even more universal directions, aligning perfectly with our empirical results that transferability improves monotonically with more auxiliary images.
>
> **3. Supplementary intuition of why adversarial mode connectivity holds: ReLU piecewise-linear structure**
>
> For ReLU-like networks, the input space is partitioned into piecewise-linear activation regions, each forming a convex polytope where the network behaves as an affine function. Although these activation regions are disjoint, misclassification regions are determined by decision boundaries rather than activation boundaries. Decision boundaries span many adjacent activation regions, producing large connected components of misclassification.
>
> Furthermore, adversarial perturbations typically lie in low-curvature directions of the loss landscape, which extend across many activation polytopes. Therefore, even if the two endpoints originate in different activation regions, they often lie within the same connected misclassification component, allowing a Bézier curve to cross region boundaries while still preserving adversarial effectiveness. This offers additional intuition for why adversarial mode connectivity is frequently observed in practice.
>
> We include this discussion in Section 3 and more details in Appendix A.2.
>
> [1]  T. Garipov, P. Izmailov, D. Podoprikhin, D. Vetrov, and A. G. Wilson, “Loss Surfaces, Mode Connectivity, and Fast Ensembling of DNNs,” 2018, arXiv. doi: 10.48550/ARXIV.1802.10026.
>
> [2]  Z. Qin et al., “Boosting the Transferability of Adversarial Attacks with Reverse Adversarial Perturbation,” 2022, arXiv. doi: 10.48550/ARXIV.2210.05968.

---

### Official Review · Reviewer_QkrL · 2025-10-28

**Soundness:** 2
**Presentation:** 3
**Contribution:** 3
**Rating:** 6
**Confidence:** 3

**Summary:**

This paper introduces MoCo-EA (Mode Connectivity Evolutionary Attack), a novel adversarial attack method that re-imagines the crossover operation in Evolutionary Algorithms (EAs). The key idea is to replace the traditional, discrete element-wise crossover with a continuous, geometry-aware Bézier crossover operator. The authors first systematically demonstrate the property of "adversarial mode connectivity," showing that successful adversarial examples for a model are not isolated but lie on connected manifolds. They then leverage this insight to develop MoCo-EA, which significantly outperforms traditional EAs in white-box settings, achieving near-perfect success rates across different \ell_p-norms while drastically reducing the number of generations, queries, and computational time required.

**Strengths:**

1. Novel and Well-Motivated Algorithmic Contribution: The core idea of replacing a discrete crossover with a continuous, optimized path is innovative and elegant. The proposed Bézier crossover is a principled way to exploit the geometry of the adversarial space, directly addressing a known weakness of traditional EAs.

2. Strong and Comprehensive Empirical Evaluation: The paper provides extensive experiments on CIFAR-10 and ImageNet across multiple l_p-norms (l_\infty,l_2,l_1) and connectivity settings (image-wise, class-wise, cross-class). The results are compelling, showing that MoCo-EA achieves universal attack success and orders-of-magnitude improvements in efficiency over a strong traditional EA baseline.

3. Excellent Presentation and Reproducibility: The paper is well-written, clearly structured, and includes detailed pseudocode, hyperparameters, and a reproducibility statement, facilitating future replication and research.

**Weaknesses:**

1.	Limited Novelty of the Core Insight: The foundational claim—that adversarial examples lie on connected manifolds—is not entirely new. While the paper does a good job of empirically validating this in the input perturbation space (as opposed to weight space), the underlying concept of mode connectivity and connected low-loss regions in neural network landscapes is a well-established area of research. The paper could more clearly position its specific contribution in this context.
2.	Unclear Practical Motivation and Positioning: The paper's motivation for focusing on evolutionary attacks in a white-box setting is somewhat underdeveloped. In a white-box scenario, where gradients are available, highly efficient and effective gradient-based attacks (e.g., PGD, AutoAttack) already exist and are the standard. The practical scenario where one would prefer a more query/computation-intensive EA, even an efficient one like MoCo-EA, is not explicitly justified. The paper would be strengthened by a clearer discussion of the niche for evolutionary attacks (e.g., for attacking non-differentiable systems or as a tool for analyzing loss landscapes) and how MoCo-EA positions itself within the broader ecosystem of adversarial attacks.
3.	Complexity vs. Benefit Trade-off: The Bézier crossover, while powerful, adds significant complexity to the EA framework, requiring inner-loop gradient-based optimization. The reviewer is left wondering if this complexity is justified given the existence of simpler, highly effective white-box methods. A discussion or experiment comparing MoCo-EA's performance and efficiency against state-of-the-art gradient-based attacks (not just traditional EAs) would help solidify its practical value.

**Questions:**

See the weaknesses.

---

> ### Author Response · Authors · 2025-11-21
> **To Reviewer QkrL (1/3)**
>
> We thank the reviewer for the constructive review. We appreciate the positive feedback on our Bézier crossover idea, our experimental results, and the clarity and reproducibility of the paper, as well as the questions about (i) the novelty of our mode-connectivity perspective, (ii) when evolutionary attacks are useful in white-box settings, and (iii) the trade-off between the added complexity of MoCo-EA and its benefits. Below, we respond to each of these points in detail.
>
> ### **Q1) The core insight that adversarial examples lie on connected manifolds is not entirely new. The underlying concept of mode connectivity and connected low-loss regions in neural network landscapes is a well-established area of research. The paper could more clearly position its specific contribution in this context.**
>
> We would like to clarify that our connectivity result concerns a fundamentally different object and is not implied by prior mode-connectivity literature. Classical mode-connectivity studies low-loss paths in weight space, not in the adversarial perturbation space [1, 2, 3]. These results do not imply that independently discovered adversarial perturbations in input space are connected, nor that continuous adversarial paths exist under strict $\ell_p$ constraints. The only work that studies connectivity directly in input space is the recent paper [4], whose setting does not involve adversarial robustness, adversarial regions, or the connectivity of adversarial perturbations. Their setting is geometrically different and cannot be used to infer the existence of continuous adversarial paths in the constrained perturbation space.
>
> In contrast, we study continuous connectivity between adversarial perturbations themselves. We show that two independently generated adversarial examples can be connected by a trajectory that (i) lies entirely within the $\ell_p$ ball and (ii) remains adversarial for every point along the curve. This demonstrates that successful adversarial perturbations form a contiguous region, rather than isolated points. Moreover, we show that entire paths and all intermediate perturbations exhibit strong transferability across images and classes, which, to our knowledge, has not been previously studied.
>
> [1] Garipov, Timur, et al. "Loss surfaces, mode connectivity, and fast ensembling of dnns." Advances in neural information processing systems 31. 2018.
>
> [2] Draxler, Felix, et al. "Essentially no barriers in neural network energy landscape." International conference on machine learning. PMLR, 2018.
>
> [3] Freeman, C. Daniel, and Joan Bruna. "Topology and Geometry of Half-Rectified Network Optimization." International Conference on Learning Representations. 2017.
>
> [4] Vrabel, Jakub, et al. "Input Space Mode Connectivity in Deep Neural Networks." The Thirteenth International Conference on Learning Representations.

---

> ### Author Response · Authors · 2025-11-21
> **To Reviewer QkrL (2/3)**
>
> ### **Q2) The motivation for using evolutionary attacks in a white-box setting is unclear, given the availability of strong gradient-based methods. The paper would be strengthened by a clearer discussion of the niche for evolutionary attacks and how MoCo-EA positions itself within the broader ecosystem of adversarial attacks.**
>
> We would like to clarify that our goal is not to replace the conventional gradient-based methods in their own regime, but to highlight the distinct role that our method plays in understanding adversarial geometry and in handling cases where gradients are unreliable or insufficient. In settings where gradients are reliable, MoCo-EA can still outperform gradient-based attacks, and when gradients are insufficient, MoCo-EA becomes crucial.
>
> First, our method reveals a global geometric structure that gradient-based attacks cannot access. Conventional gradient-based attacks follow a single optimization trajectory, which restricts them to a narrow region of the loss landscape and often collapses them into a single adversarial direction. In contrast, MoCo-EA operates on entire populations of perturbations. This allows it to explore multiple regions of the perturbation space simultaneously, reveal global structure such as connected adversarial manifolds, uncover wide contiguous regions where adversarial perturbations remain transferable, and avoid the mode-collapse behavior commonly seen in gradient-based updates. These insights are not accessible to PGD or AutoAttack, whose purpose is to find one strong perturbation, not to characterize the geometry of adversarial regions. MoCo-EA exploits this population-level perspective to systematically analyze adversarial path-connectivity, an insight that conventional gradient-based attacks cannot provide.
>
> Second, MoCo-EA remains essential in settings with unreliable or unstable gradients, e.g., in scenarios with obfuscated gradients [5]. These settings make conventional gradient-based attacks far less reliable and sometimes ineffective. Evolutionary algorithms, by contrast, depend partially on loss evaluations and remain fully applicable. MoCo-EA can therefore attack models or pipelines that conventional gradient-based methods struggle with.
>
> To demonstrate this, we evaluate MoCo-EA on (1) a robustly trained CIFAR-10 ResNet-50 model, and (2) models with obfuscated gradients. On the robust model, MoCo-EA achieves higher success rates than strong white-box baselines.
>
> | Norm           | PGD | MIFGSM | AutoAttack | AdaptiveAutoAttack | MoCo-EA |
> |----------------|-----|--------|------------|---------------------|---------|
> | $\ell_\infty$  | 45  | 45     | 46         | 45                  | 48      |
>
> Under obfuscated-gradient settings, gradient-based approaches become far less reliable, whereas MoCo-EA remains highly effective.
>
> | Norm           | PGD | MIFGSM | AutoAttack | AdaptiveAutoAttack | MoCo-EA |
> |----------------|-----|--------|------------|---------------------|---------|
> | $\ell_\infty$  | 17  | 17     | 17         | 16                  | 32      |
>
>
> We include this discussion in Section 5.3 and report the experimental results in Table 6.
>
> [5] Athalye, Anish, Nicholas Carlini, and David Wagner. "Obfuscated gradients give a false sense of security: Circumventing defenses to adversarial examples." International conference on machine learning. PMLR, 2018.

---

> ### Author Response · Authors · 2025-11-21
> **To Reviewer QkrL (3/3)**
>
> ### **Q3) The Bézier crossover adds significant complexity to the EA framework, requiring inner-loop gradient-based optimization. It is unclear whether this is justified given simpler, strong white-box methods. A discussion or experiment comparing MoCo-EA's performance and efficiency against state-of-the-art gradient-based attacks would help solidify its practical value.**
>
> We want to clarify that the added complexity is minimal, as the Bézier crossover requires only a few inner-loop gradient steps (n = 5), yet it brings substantial gains. As shown in Table 5, MoCo-EA converges much faster, succeeding in far fewer generations, achieving a 93–99% reduction across all norms.  It also requires far fewer queries and lower runtime while still reaching 100% success rates across all norms, clearly justifying the small increase in complexity.
>
> To further address the reviewer’s request for a comparison with state-of-the-art gradient-based attacks, we reported the runtime of AutoAttack in the table below. Although AutoAttack is faster, gradient-based attacks inherently struggle in settings where gradients are unreliable, and MoCo-EA also outperforms even when gradients are sufficient.  These experiments are discussed in our response to Q2. Thus, the added complexity of the Bézier crossover is minor relative to the robustness and performance benefits it provides. We also include a computational complexity analysis of Bézier crossover in Appendix A.6.
>
> | Norm        | Traditional EA        | AutoAttack       | MoCo-EA          |
> |-------------|------------------------|-------------------|-------------------|
> | $\ell_\infty$ | 95.14 ± 60.22          | 0.58 ± 0.29        | 5.05 ± 0.04        |
> | $\ell_2$      | 152.15 ± 58.07         | 0.54 ± 0.00        | 5.01 ± 0.02        |
> | $\ell_1$      | 118.19 ± 81.81         | 0.72 ± 0.00        | 4.61 ± 1.48        |

---

### Official Review · Reviewer_v25a · 2025-10-31

**Soundness:** 3
**Presentation:** 4
**Contribution:** 2
**Rating:** 4
**Confidence:** 3

**Summary:**

The paper considers optimisation of adversarial attacks using evolutionary algorithms. The main claimed contributions of are

1. A new crossover operator that produces an offspring attack by sampling uniformly on a Bezier curve between the two parent attacks. They present an evolutionary algorithm (MoCo-EA) which applies this operator and compare it with a baseline evolutionary algorithm.
2. Showing that the set of successful attacks can form a contiguous region in parameter space, rather than scattered points.

**Strengths:**

The paper reports results from a substantial number of experiments on established benchmarks, including CIFAR-10 and ImageNet. The domain specific crossover-operator proposed outperforms established crossover operators.

**Weaknesses:**

Adversarial attacks is not my area of research, however I do not feel the contributions are sufficiently significant.

With regards to the first contribution, I am missing a more comprehensive comparison with state of the art methods. The paper compares with a "baseline evolutionary algorithm", but I would expect to see more specific details about this baseline algorithm, and also comparison with the best methods. Values are reported with mean and standard deviations. It would have been nice to see results of statistical tests.

I am unsure about the novelty of the second contribution. E.g., the paper by Tramer et al (2017) (https://arxiv.org/abs/1704.03453) seems to indicate a similar phenomenon. Given that evolutionary algorithms are effective in finding attacks, and that EAs cannot solve needle in the haystack problems, it seems intuitive that attacks form contiguous spaces in the parameter space.

**Questions:**

Please comment on the novelty of your second contribution (see my comments above), taking into account papers such as https://arxiv.org/abs/1704.03453.

---

> ### Author Response · Authors · 2025-11-21
> **To Reviewer v25a (1/2)**
>
> We appreciate the reviewer’s positive feedback on our experiments and the effectiveness of our domain-specific crossover operator. Moreover, we want to highlight our several contributions: (1) we identify and validate a continuous connectivity structure of adversarial perturbations within the input space, (2) we show that intermediate points along optimized paths achieve higher transferability than endpoints, (3) we present a simple path-based framework to analyze this geometry, and (4) we develop MoCo-EA, a crossover operator that leverages this structure on top of the standard EA pipeline. In addition, (5) we introduce three settings (image-wise, class-wise, and cross-class) to study adversarial mode connectivity under increasing levels of generalization. We also thank the reviewer for pointing out the need for broader comparisons, clearer baseline details, additional statistical tests, and concerns regarding the novelty. We address these concerns in detail below.
>
> ### **Q1) Regarding the first contribution, a more comprehensive comparison with state-of-the-art methods appears to be missing.**
>
> We evaluate MoCo-EA under two experimental settings with PGD [1] and AutoAttack [2], which are widely used gradient-based adversarial attacks for robustness evaluation [3, 4, 5]. We then additionally compare MoCo-EA with other state-of-the-art attacks such as MI-FGSM [6] and Adaptive AutoAttack [4]. For MI-FGSM [6] and Adaptive AutoAttack [4], we use their official implementations. The original codebases do not provide $\ell_2$ or $\ell_1$ versions (and Adaptive AutoAttack does not include an $\ell_1$ implementation), so we do not report those settings.
>
> First, we compare attack success rates on an adversarially trained CIFAR-10 ResNet-50 model.  As shown below, MoCo-EA achieves higher success rates than these strong baselines, indicating that our method provides a more effective attack compared to other white-box attack baselines.
>
> | Norm           | PGD | MIFGSM | AutoAttack | AdaptiveAutoAttack | MoCo-EA |
> |----------------|-----|--------|------------|---------------------|---------|
> | $\ell_\infty$  | 45  | 45     | 46         | 45                  | 48      |
>
> Second, we evaluate under obfuscated gradient settings [7] on a standard model. In these scenarios, gradient-based attacks often become unreliable or even ineffective. In contrast, evolutionary algorithms rely partially on loss evaluations and remain fully applicable. As shown below, MoCo-EA outperforms gradient-based methods under these challenging conditions.
>
> | Norm           | PGD | MIFGSM | AutoAttack | AdaptiveAutoAttack | MoCo-EA |
> |----------------|-----|--------|------------|---------------------|---------|
> | $\ell_\infty$  | 17  | 17     | 17         | 16                  | 32      |
>
> These results are included in Section 5.3 and Table 6. For further details, please refer to that section.
>
>
> [1] Madry, Aleksander, et al. "Towards Deep Learning Models Resistant to Adversarial Attacks." International Conference on Learning Representations. 2018.
>
> [2] Croce, Francesco, and Matthias Hein. "Reliable evaluation of adversarial robustness with an ensemble of diverse parameter-free attacks." International conference on machine learning. PMLR, 2020.
>
> [3] Zhang, Xu, et al. "Optimizing Robustness and Accuracy in Mixture of Experts: A Dual-Model Approach." Forty-second International Conference on Machine Learning.
>
> [4] Liu, Ye, et al. "Practical evaluation of adversarial robustness via adaptive auto attack." Proceedings of the IEEE/CVF Conference on Computer Vision and Pattern Recognition. 2022.
>
> [5] Croce, Francesco, and Matthias Hein. "Reliable evaluation of adversarial robustness with an ensemble of diverse parameter-free attacks." International conference on machine learning. PMLR, 2020.
>
> [6] Dong, Yinpeng, et al. "Boosting adversarial attacks with momentum." Proceedings of the IEEE conference on computer vision and pattern recognition. 2018.
>
> [7]  Athalye, Anish, Nicholas Carlini, and David Wagner. "Obfuscated gradients give a false sense of security: Circumventing defenses to adversarial examples." International conference on machine learning. PMLR, 2018.

---

> ### Author Response · Authors · 2025-11-21
> **To Reviewer v25a (2/2)**
>
> ### **Q2) The paper compares its method with a “baseline evolutionary algorithm,” but more specific details about this baseline would be expected.**
>
> The “traditional EA” is a standard evolutionary attack that follows a standard population-based procedure. It initializes a population of random perturbations inside the same $\ell_p$ budget, scores them by attack success, and iteratively updates the population using tournament selection, element-wise crossover, Gaussian mutation, and top-k preservation across generations. MoCo-EA keeps this evolutionary pipeline unchanged and differs only in the crossover step, where it replaces element-wise crossover with our geometry-aware Bézier crossover. We refer to this unmodified version as the “traditional EA” baseline. We have updated Section 5.3 to clarify this setup and provide additional implementation details in Appendix A.4.
>
> ### **Q3) Values are reported with means and standard deviations, but it would have been helpful to also include statistical tests.**
>
> We conduct one-sided paired t-tests on ImageNet results in Table 5 to evaluate whether the improvements of MoCo-EA over the traditional evolutionary attack are statistically significant. For success rates, we additionally repeated the experiments with 5 different random seeds and report the mean ± standard deviation across seeds. Results show that for success rates, the p-values are all much smaller than 0.05, which means the differences are statistically significant. For the number of generations, the p-values indicate statistically significant differences for ℓ∞ and ℓ2 (p < 0.05), while ℓ1 still shows a clear improving trend in favor of MoCo-EA. We include these statistical results in Appendix A.6 and Table 10.
>
> | Norm           | Metric       | Traditional EA (mean ± std) | p-value   |
> |----------------|--------------|------------------------------|-----------|
> | $\ell_\infty$  | Succ. rate   | 87.3 ± 2.8                   | 2.65E-04  |
> | $\ell_\infty$  | Generations  | 456.8 ± 309.0                | 9.06E-08  |
> | $\ell_2$       | Succ. rate   | 12.0 ± 1.8                   | 2.22E-08  |
> | $\ell_2$       | Generations  | 24.8 ± 9.8                   | 1.23E-02  |
> | $\ell_1$       | Succ. rate   | 35.3 ± 4.5                   | 2.73E-06  |
> | $\ell_1$       | Generations  | 13.3 ± 22.6                  | 6.29E-02  |
>
>
> ### **Q4) Unsure about the novelty of the second contribution. The paper by Tramer et al (2017) seems to indicate a similar phenomenon.**
>
> We clarify that the findings of [8] are fundamentally different from ours. Their work demonstrates that different models trained on the same task share a large, high-dimensional adversarial subspace, which explains why single-point adversarial perturbations often transfer across classifiers. However, they (1) do not investigate the geometry of adversarial regions in the input space for a single model; (2) do not construct continuous paths between adversarial perturbations nor analyze connectivity inside the $\ell_p$ ball; and (3) do not analyze transferability along such paths. Thus, while [8] explain shared subspaces between models, their results do not address nor imply connectivity of adversarial perturbations within the input space of one model.
>
> In contrast, our contribution studies an orthogonal question: Does the set of adversarial perturbations for one model/input form a continuous manifold inside the $\ell_p$ ball? We show that such paths not only exist but also exhibit higher transferability at intermediate points, revealing that successful attacks occupy a connected low-loss region rather than isolated points. To our knowledge, this is the first work to characterize and empirically validate input-space adversarial connectivity.
>
> [8] Tramèr, Florian, et al. "The space of transferable adversarial examples." arXiv preprint arXiv:1704.03453 (2017).
>
> ### **Q5) Given that evolutionary algorithms are effective in finding attacks, and that EAs cannot solve needle in the haystack problems, it seems intuitive that attacks form contiguous spaces in the parameter space.**
>
> We would like to clarify that our connectivity result is a scientific observation about the geometry of adversarial perturbations. It is completely independent of the evolutionary algorithm we later build on top of it. MoCo-EA exploits this geometry, but the discovery itself is not derived from EA behavior nor from prior EA literature. Moreover, the presence of multiple adversarial perturbations does not imply that they are connected. No prior EA-based work constructs, optimizes, or proves a continuous adversarial path between perturbations. Therefore, EA behavior is not evidence for input-space connectivity.

---

### Author Response · Authors · 2025-11-21
**General Response**

Dear Reviewers, AC, SAC, and PC,

Thank you for the time and effort you dedicated to reviewing our manuscript.

Our paper introduces a Bézier crossover framework that replaces traditional discrete crossover with a continuous, optimized path between parent perturbations. We appreciate the reviewers’ recognition of both the novelty of this idea and the rigor of our empirical evaluation. Below, we summarize the main strengths highlighted by the reviewers, along with the revisions made in response to their comments.

### **Key strengths noted by the reviewers:**
1. **Comprehensive evaluation.** Extensive experiments on CIFAR-10 and ImageNet across  $\ell_\infty$, $\ell_2$, and $\ell_1$ norms, and across multiple connectivity settings (image-wise, class-wise, cross-class). (Reviewers v25a, QkrL)
2. **Strong performance.** MoCo-EA consistently outperforms baselines across all settings. (All reviewers)
3. **Novel and well-motivated idea.** The continuous Bézier-based crossover provides a structured, gradient-informed interpolation between parent perturbations, improving diversity, transferability, and robustness compared to discrete crossover. (Reviewers QkrL, D1Af)
4. **Clarity and reproducibility.** The paper is clearly written, well-organized, and includes detailed pseudocode and hyperparameters, facilitating reproducibility. (Reviewer QkrL)
5. **Insightful motivation.** MoCo-EA addresses a limitation of gradient-based attacks—their restriction to local regions of the loss landscape—and motivates the study of continuous connectivity between adversarial perturbations. (Reviewer D1Af)

### **Key revisions in response to the comments:**
We have carefully revised the manuscript, and the main updates are summarized below:

1. **Additional baselines and comparisons.** Added more white-box attack baselines, including additional results and discussion in Section 5.3 and Table 6. (Reviewers v25a, QkrL)
2. **Clarification of the baseline evolutionary algorithm.** Clarified what the “baseline evolutionary algorithm” refers to in Section 5.3 and provided further details in Appendix A.4.   (Reviewer v25a)
3. **Statistical tests.** Added one-sided paired t-tests in Appendix A.6 and Table 10. (Reviewer v25a)
4. **Computational complexity analysis.** Included a computational complexity analysis of the Bézier crossover in Appendix A.6. (Reviewers QkrL, D1Af)
5. **Additional intuition and motivation.** Expanded the explanation in Section 3 and added more details in Appendix A.2.

All the revised content is highlighted in blue.

---

### Note · Program_Chairs · 2026-01-17
**Submission Desk Rejected by Program Chairs**

The following references in this submission do not refer to real documents and/or have major errors in bibliographic information:

 Sixiao Zhao, Zixuan Liu, Ji Lin, and Song Han. Bridging mode connectivity in loss landscape and function space. In Advances in Neural Information Processing Systems (NeurIPS), 2020.